# Structural basis for human NKCC1 inhibition by loop diuretic drugs

Yongxiang Zhao [1,2], Pietro Vidossich[3], Biff Forbush[4], Junfeng Ma [5], Jesse Rinehart [4,6], Marco De Vivo[3] & Erhu Cao [1✉]

## Abstract

$Na^+–K^+–Cl^-$ cotransporters functions as an anion importers, regulating trans-epithelial chloride secretion, cell volume, and renal salt reabsorption. Loop diuretics, including furosemide, bumetanide, and torsemide, antagonize both NKCC1 and NKCC2, and are first-line medicines for the treatment of edema and hypertension. NKCC1 activation by the molecular crowding sensing WNK kinases is critical if cells are to combat shrinkage during hypertonic stress; however, how phosphorylation accelerates NKCC1 ion transport remains unclear. Here, we present co-structures of phospho-activated NKCC1 bound with furosemide, bumetanide, or torsemide showing that furosemide and bumetanide utilize a carboxyl group to coordinate and co-occlude a $K^+$, whereas torsemide encroaches and expels the $K^+$ from the site. We also found that an amino-terminal segment of NKCC1, once phosphorylated, interacts with the carboxyl-terminal domain, and together, they engage with intracellular ion exit and appear to be poised to facilitate rapid ion translocation. Together, these findings enhance our understanding of NKCC-mediated epithelial ion transport and the molecular mechanisms of its inhibition by loop diuretics.

**Keywords** $Na^+–K^+–Cl^-$ Cotransporter; Loop Diuretics; ATP Regulatory Site; With-no-lysine Kinase
**Subject Categories** Pharmacology & Drug Discovery; Structural Biology; Urogenital System

## Introduction

Loop diuretic drugs have been essential medications widely prescribed for the management of edema and hypertension for more than 6 decades (Baletic et al, 2022; Elliott and Jurca, 2016; Malha and Mann, 2016; Sica et al, 2011; Voelker et al, 1987). Furosemide, torsemide, and bumetanide are the top 3 most commonly used loop diuretics in the United States with a combined prescription of ~40 million each year. Loop diuretics inhibit both the renal-specific $Na^+–K^+–Cl^-$ cotransporter 2 (NKCC2) and ubiquitously expressed paralogous transporter NKCC1 with almost equal potency (Orlov et al, 2015). The NKCC2 antagonism blunts salt and water retention by the kidneys and accounts for the diuretic and anti-hypertensive effects of loop diuretic drugs (Shankar and Brater, 2003; Wittner et al, 1991). Inhibition of NKCC1, on the other hand, can cause the side-effect of ototoxicity (Ding et al, 2016; Rybak, 1993) as NKCC1 contributes to the maintenance of high extracellular $[K^+]$ of endolymph in inner ear by uptake of $K^+$ at the basolateral membrane for subsequent secretion via $K^+$ channels in the apical membrane (Delpire et al, 1999; Flagella et al, 1999). Besides a promiscuous action on both NKCCs, loop diuretics (except for ethacrynic acid) also bear a sulfur atom and are thus contraindicated in patients who are allergic to sulfur-containing compounds (Juang et al, 2006; Oh and Han, 2015; Slatore and Tilles, 2004). We and others recently reported co-structures of human NKCC1 complexed with bumetanide, demonstrating that bumetanide binds to and occludes the extracellular ion entryway of NKCC1 (Moseng et al, 2022; Zhao et al, 2022a). Our NKCC1/bumetanide co-structure also clearly showed that bumetanide's carboxyl group directly coordinates and co-occludes a $K^+$ ion in its original binding site (Zhao et al, 2022a), explaining why bumetanide binding requires $K^+$ (Borgogno et al, 2021; Savardi et al, 2021) and why modifications of bumetanide's carboxyl group lead to loss of diuretic activity in a previous structure-activity relationship study (Lykke et al, 2016). However, we are yet to elucidate the sites and mechanisms of action of furosemide and torsemide which are the top two most prescribed loop diuretic drugs in the United States. A recent study showed that furosemide (and bumetanide) bind to an intracellular site, but this finding has since been questioned (Flygaard et al, 2023; Moseng et al, 2023; Moseng et al, 2022). Perhaps the most significant knowledge gap in loop diuretics pharmacology is how torsemide, which lacks the $K^+$-coordinating carboxyl group found in bumetanide and furosemide, binds to and inhibits the two NKCC transporters.

NKCC1 functions as a key $Cl^-$ importer and maintains intracellular $[Cl^-]$ in neurons (Kaila et al, 2014). NKCC1 thus represents an attractive therapeutic target for the treatment of

---

[1]Department of Biochemistry, University of Utah School of Medicine, Salt Lake City, UT 84112-5650, USA. [2]Key Laboratory of Magnetic Resonance in Biological Systems, State Key Laboratory of Magnetic Resonance and Atomic and Molecular Physics, National Center for Magnetic Resonance in Wuhan, Wuhan Institute of Physics and Mathematics, Innovation Academy for Precision Measurement Science and Technology, Chinese Academy of Sciences, 430071 Wuhan, P. R. China. [3]Laboratory of Molecular Modelling & Drug Discovery, Istituto Italiano di Tecnologia, GenoaVia Morego 30, 16163, Italy. [4]Department of Cellular and Molecular Physiology, Yale University School of Medicine, New Haven, CT, USA. [5]Lombardi Comprehensive Cancer Center, Georgetown University Medical Center, Washington DC 20057, USA. [6]Systems Biology Institute, Yale University, West Haven, CT, USA. ✉E-mail: erhu.cao@biochem.utah.edu

various brain disorders where aberrantly elevated NKCC1 activity and intracellular [Cl$^-$] in neurons lead to attenuated inhibitory synaptic transmission that critically depends on an inward directed [Cl$^-$] electrochemical gradient (Kaila et al, 2014; Savardi et al, 2021). Bumetanide has been repurposed to inhibit NKCC1 in the brain in several open-label trials but with limited success in large part because bumetanide bears a carboxyl group and barely crosses the blood-brain barrier (Loscher and Kaila, 2022; Savardi et al, 2021). In this regard, carboxyl group free loop diuretics such as torsemide could serve as better starting compounds for the development of brain-penetrable NKCC1 inhibitors. NKCC1/torsemide co-structures would provide a sorely needed blueprint to guide the medicinal chemistry efforts aimed to design new torsemide derivatives with improved brain penetrance and specificity for NKCC1.

Another important mechanistic aspect of NKCC1 is that its ion transport activity is profoundly regulated by (de)phosphorylation. NKCC1, in concert with K$^+$-Cl$^-$ cotransporters (KCCs) which are cousins of NKCCs but serve as Cl$^-$ extruders, contribute to the maintenance of cell volume by importing or extruding ions (and obligatory water) downstream of the volume-sensing WNKs-SPAK kinase cascade (de Los Heros et al, 2018). Cell shrinkage induces molecular crowding and activates the WNKs-SPAK kinase signaling cascade, which in turn phosphorylates NKCC1 and KCCs, resulting in activation of the former but inhibition of the latter and, consequently, an influx of ion and water to restore cell volume (Boyd-Shiwarski et al, 2022; de Los Heros et al, 2018). In contrast, when cells are under hypotonic stress, WNKs-SPAK and NKCC1 activities are suppressed and a net ion efflux via dephosphorylated and active KCCs defends against cell swelling. Although phosphoacceptor residues of NKCC1 have been identified within its cytoplasmic N-terminal domain (Darman and Forbush, 2002), it remains mysterious whether/how (de)phosphorylation of these residues triggers conformational changes that then propagate to ultimately impact the ion translocation pathway housed within the transmembrane domains. Alternatively, these phosphoacceptor residues, once phosphorylated by WNKs-SPAK, could serve as docking sites for as-yet unidentified cellular factors that in turn accelerate NKCC1 ion transport activity.

Here we determined three cryo-EM structures of phosphorylated NKCC1 individually bound with furosemide, torsemide, and bumetanide (Fig. 1; Appendix Table S1), unlocking two modes of interaction of loop diuretics with NKCC1. Furosemide and bumetanide exhibit a K$^+$-dependent antagonism because they both bear and use a carboxyl group to coordinate and occlude a K$^+$ ion within a pocket at the entryway of the extracellular ion translocation path of NKCC1. These results clarify previous debates on their proposed modes of action (Flygaard et al, 2023; Moseng et al, 2023; Moseng et al, 2022), which we showed are analogous in targeting the same receptor site in the transmembrane domains (TMDs) of NKCC1. We also revealed that torsemide, which lacks the carboxyl group, encroaches into and expels K$^+$ from its binding site. In addition, our structures, determined at 2.5 Å–2.7 Å resolution, showed that three threonine residues, which are located at the cytoplasmic N-terminal domain of NKCC1, each bear a PO$_4^{3-}$ group that are seen to interact with positively charged arginine and lysine residues on C-terminal domain. Once N- and C-terminal domains tightly associate with each other upon phosphorylation, they interact with the intracellular ion exit and

appear to stabilize NKCC1 in a conformation that is conducive to rapid ion translocation. Our study could catalyze computational discovery of new small molecules or protein biologics to target NKCCs ion translocation path and phosphoregulatory interfaces as next generation of diuretic drugs with greater specificity and with reduced side effects.

# Results

## Overall architecture of phospho-activated NKCC1 in complex with furosemide, torsemide, and bumetanide

Furosemide, bumetanide, and torsemide inhibit NKCC transporters with half maximum inhibitory concentrations (IC$_{50}$s) within a low micromolar range (Hampel et al, 2018) (Figs. 2E and EV1A,B), hindering preparation of stable NKCC1/drug complexes for structural analyses. Since loop diuretics bind to active NKCC1 with enhanced affinities upon stimulation of the upstream WNKs-SPAK kinases (Haas and Forbush, 1986; Lytle and Forbush, 1992), we sought to determine co-structures of phospho-activated NKCC1 in complex with furosemide, bumetanide, and torsemide. For this effort, we generated a HEK293 cell line that stably co-expresses NKCC1 together with a constitutively active WNK1 kinase (residues 1-483 harboring S378D activating mutation) (Schiapparelli et al, 2021), SPAK kinase (Dowd and Forbush, 2003), and Mo25 (a co-activator of SPAK) (Filippi et al, 2011) (Fig. EV1C,D). We additionally treated the cells in a Cl$^-$ and K$^+$ free buffer that is known to activate WNKs (Goldsmith and Rodan, 2023; Richardson et al, 2008), as well as with calyculin (a phosphatase inhibitor) to blockade NKCC1 dephosphorylation by endogenous phosphatases (Flemmer et al, 2002; Ishihara et al, 1989). This integrated approach of reconstituting the WNK1-SPAK-NKCC1 pathway in HEK293 cells coupled with maintaining activating conditions in all steps enabled purification of NKCC1 in a phospho-activated state as all key phosphoacceptors (e.g., Thr212, Thr217, and Thr230) were verified by mass spectrometry to each bear a phosphate group (Appendix Fig. S1; referred to as pNKCC1 hereafter). We then determined pNKCC1/furosemide, pNKCC1/torsemide, and pNKCC1/bumetanide co-structures at 2.7 Å, 2.6 Å, and 2.5 Å resolution, respectively (Appendix Figs. S2–S8).

In all the three co-structures, NKCC1 assembled as a C2-symmetric dimer in which two TMDs are separated by ~30 Å within the lipid bilayer and two cytosolic C-terminal domains (CTDs) interdigitate and associate extensively with the TMDs from beneath (Fig. 1). An N-terminal domain (NTD; residues Thr203-Leu250) was seen to bear phosphorylated Thr212 (pT212), Thr217 (pT217), and Thr230 (pT230), which help to anchor the NTD to a concave surface of CTD via PO$_4^{3-}$-mediated electrostatic interactions. NTD/CTD of one NKCC1 subunit crosses the two-fold axis and interacts with the TMD of a second subunit (Fig. 1). This domain-swapped organization pairs the ion translocation path of one NKCC1 subunit with the phosphoregulatory domains of another subunit, dictating that NKCC1 must function as an obligatory dimer. Our structures also showed that furosemide, torsemide, and bumetanide all nestle within an orthosteric pocket in an extracellular vestibule of NKCC1 ion translocation path that would otherwise coordinate K$^+$ and Cl$^-$ ions (Appendix Fig. S9). In doing so, these three diuretic drugs occlude the extracellular ion

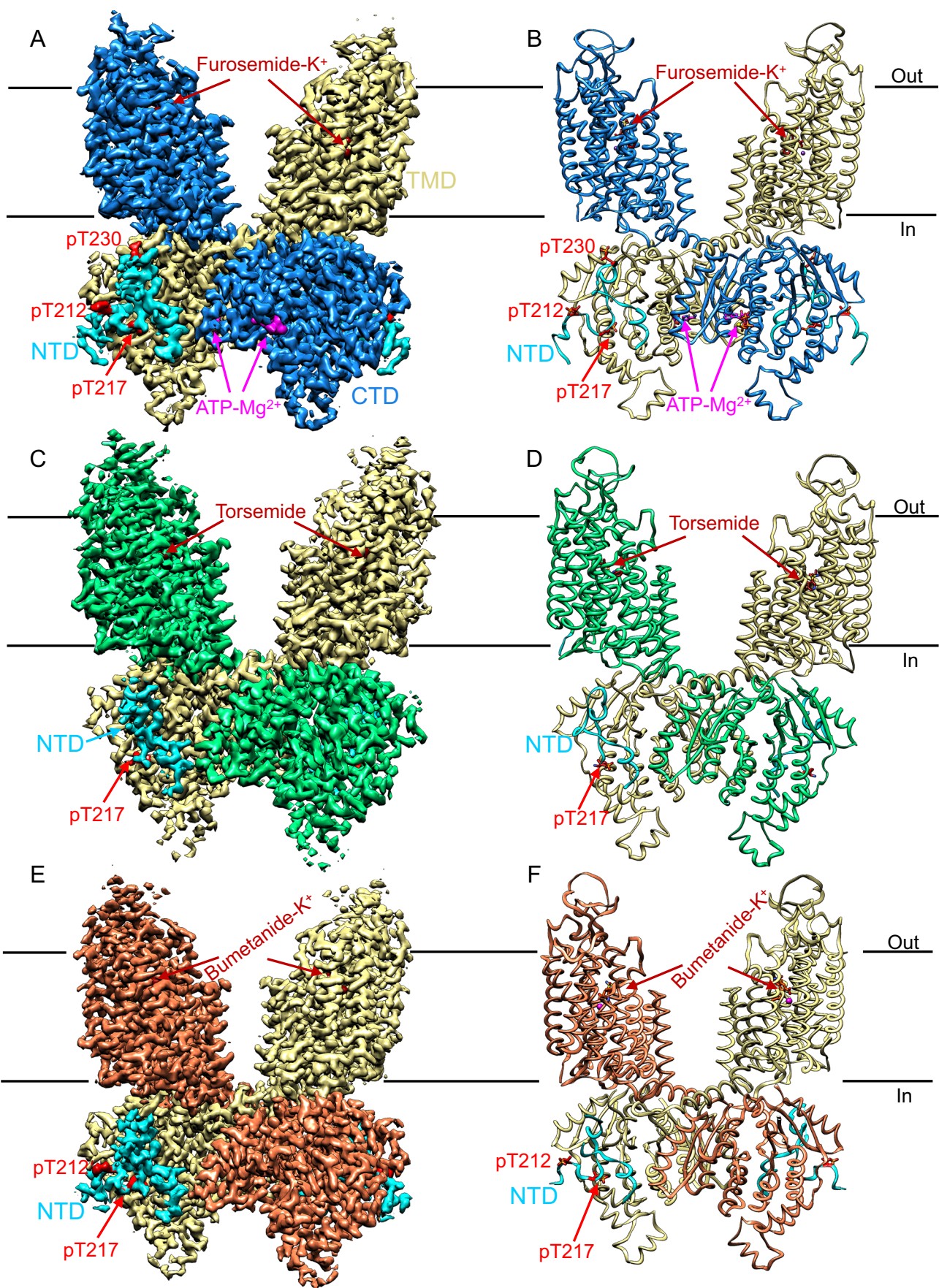

**Figure 1. Structures of phospho-activated human NKCC1 in complex with furosemide, torsemide, and bumetanide.**

(A, B) Overall structure of pNKCC1 bound with furosemide shown in map (A) and ribbon diagram (B). (C, D) Overall structure of pNKCC1 bound with torsemide shown in map (C) and ribbon diagram (D). (E, F) Overall structure of pNKCC1 bound with furosemide shown in map (E) and ribbon diagram (F). Data information: Subunit A of NKCC1 is colored-coded dodger blue, or spring green, or coral, while subunit B is colored-coded khaki in all three co-structures. ATP, phosphoacceptors, furosemide, torsemide, and bumetanide are highlighted. The N-terminal phosphoregulatory region is shown in cyan with the phosphorylation sites highlighted in red.

translocation path and arrest NKCC1 in an almost identical outward-open conformation along its transport cycle. In our NKCC1/furosemide co-structure determined at 2.6 Å resolution, an adenosine triphosphate (ATP)-Mg$^{2+}$ molecule was clearly seen in an amphipathic pocket in the CTD (Fig. 3A), which contradicts with the assignment of this pocket as a receptor site for bumetanide and furosemide (Moseng et al, 2022). Overall, our three co-structures highlight the orthosteric loop diuretics binding site, intracellular ATP-binding pocket, and WNKs-SPAK catalyzed phosphoregulatory interfaces as promising druggable sites for the development of novel therapeutics to treat various disorders caused by deranged NKCCs activity (Koumangoye et al, 2021; Savardi et al, 2021).

## Two modes of NKCC1 antagonism by loop diuretic drugs

Furosemide, torsemide, and bumetanide all intercalated into an extracellular vestibule of NKCC1 ion translocation path, arresting NKCC1 in an outward-open conformation along its transport cycle (Figs. 2A,C and EV2 and 3). The loop diuretics binding pocket is unobstructed from the extracellular side, explaining why loop diuretics, when administered externally to cells, can rapidly inhibit NKCC1 within 30 s in cell-based ion flux assay (Neumann et al, 2022; Ruiz Munevar et al, 2024; Zhao et al, 2022a). Our structures also agree with previous mutagenesis studies that have delineated the bumetanide receptor site within the TMDs of NKCC1 (Isenring and Forbush, 2001; Isenring and Forbush, 1997). Furosemide, torsemide, and bumetanide all act as orthosteric inhibitors of NKCCs as they share a binding pocket with the substrate ions (Appendix Fig. S9). In particular, furosemide, torsemide, and bumetanide all clash with the Cl$^-$ ion in the so-called Cl$^-$ site 1, but co-occlude a Cl$^-$ ion in site 2, consistent with the biphasic effect of Cl$^-$ on bumetanide binding such that low [Cl$^-$] up to 10 mM concentration enhances bumetanide binding but then progressively diminishes bumetanide affinity as [Cl$^-$] continues to increase (Forbush and Palfrey, 1983). Since bumetanide binding has been described in our prior work (Zhao et al, 2022a), we here focus on sites and mechanisms of action of furosemide and torsemide.

Furosemide intercalates into the NKCC1 extracellular ion entryway and assumes an extended configuration, with its sulfamoyl group facing the extracellular side and its furfuryl amine group reaching almost the midway of membrane (Fig. 2A). We confirmed that furosemide maintains its experimental pose, albeit only together with ions, during 250 ns molecular dynamics (MD) simulations (Fig. EV4A–C). Notably, furosemide's sulfamoylbenzoic moiety, which represents the molecular scaffold of carboxyl group-bearing loop diuretics, interacts extensively with residues in TM1, TM3, TM6, and TM8 (Fig. 2B). Here, the sulfamoyl group engages with Ile493 on TM6a and Arg307 on TM1b, plausibly explaining why ethacrynic acid that lacks this group is significantly less potent than sulfonamide loop diuretics (Hampel et al, 2018). Beneath this top layer of

interactions, furosemide's chlorine and carboxyl group project toward opposite sides of the ion translocation path and interact with residues residing on TM1b, TM3, and TM6a. Here, the chlorine atom barely contacts NKCC1, while the carboxyl group forms hydrogen bonds with Val302 and Met303 located on TM1b, and Ala497 residing on TM6a. The carboxyl group expels the Cl$^-$ ion from the so-called the Cl$^-$ site 1 and establishes ionic interaction with the K$^+$ ion that was clearly seen to be co-occluded in the structure (Fig. EV3A; Appendix Fig. S9). Further toward the midway of ion translocation path, furosemide's furfuryl amine group is buried within a hydrophobic cavity and interacts with Met382 protruding from TM3, and Ile678, Ser679, and Phe682 which are all located on TM8. We previously showed that bumetanide analogously uses its carboxyl group to coordinate and co-occlude a K$^+$ ion as does furosemide (Zhao et al, 2022a), hinting at a role for extracellular K$^+$ in tuning potency of carboxyl group bearing loop diuretics. Indeed, we found that furosemide exhibits threefold higher potency in a K$^+$ buffer than in a K$^+$-free buffer (Fig. 2F).

Torsemide lacks the carboxyl group which is indispensable for furosemide and bumetanide to co-occlude a K$^+$ ion and contributes to their K$^+$-dependent antagonism on NKCCs. Our NKCC1/torsemide co-structure showed that torsemide adopts a U-shape configuration with its sulfonylpyridine group, which marks the U-turn, facing TM1b and TM6a on one side and the propan-2-ylurea and methylanilino groups projecting toward TM3 and TM8 on the other side (Figs. 2C,D and EV3C). Torsemide notably reaches slightly deeper into the NKCC1 ion translocation path than do furosemide and bumetanide, and in doing so, it completely occupies and expels ions from the Cl$^-$ site 1 and K$^+$ site (Appendix Fig. S9). Just beneath the extracellular ion entryway, torsemide's propan-2-ylurea group forms salt bridge with the main chain nitrogen atom of Val302 in TM1a and also engages with Gly301, Val302, and Met303 in TM1a and Met382 in TM3 via hydrophobic interactions. At the U-turn, torsemide's sulfonylpyridine group establishes hydrophobic interactions with Ala497 and Pro496 in the TM6b. Further toward the midway of NKCC1 ion translocation path, torsemide's methylanilino group projects toward a hydrophobic pocket and interacts with Tyr383 in TM3, and Ile677 and Ser678 in TM8. Torsemide can exist in two neutral forms at pH 7.4 (Fig. EV4D): one form with a protonated pyridine and deprotonated sulfonylurea (Tor_NH_N) and another with a neutral pyridine and neutral sulfonylurea (Tor_N_NH). We thus explored the favored state of torsemide when bound to NKCC1 by MD simulations. Our MD results showed that the favored (i.e., more stable) state of torsemide is the one that features a protonated and positively charged pyridine (Tor_NH_N, Fig. EV4E–G). In this state, torsemide's pyridine can replace the K$^+$ ion in its binding site, while its deprotonated and negatively charged sulfonylurea can substitute the Cl$^-$ ion at the so-called site 1. Our MD simulations hint that the protonation state (and consequently potency) of torsemide could be affected by the acidity of body fluids.

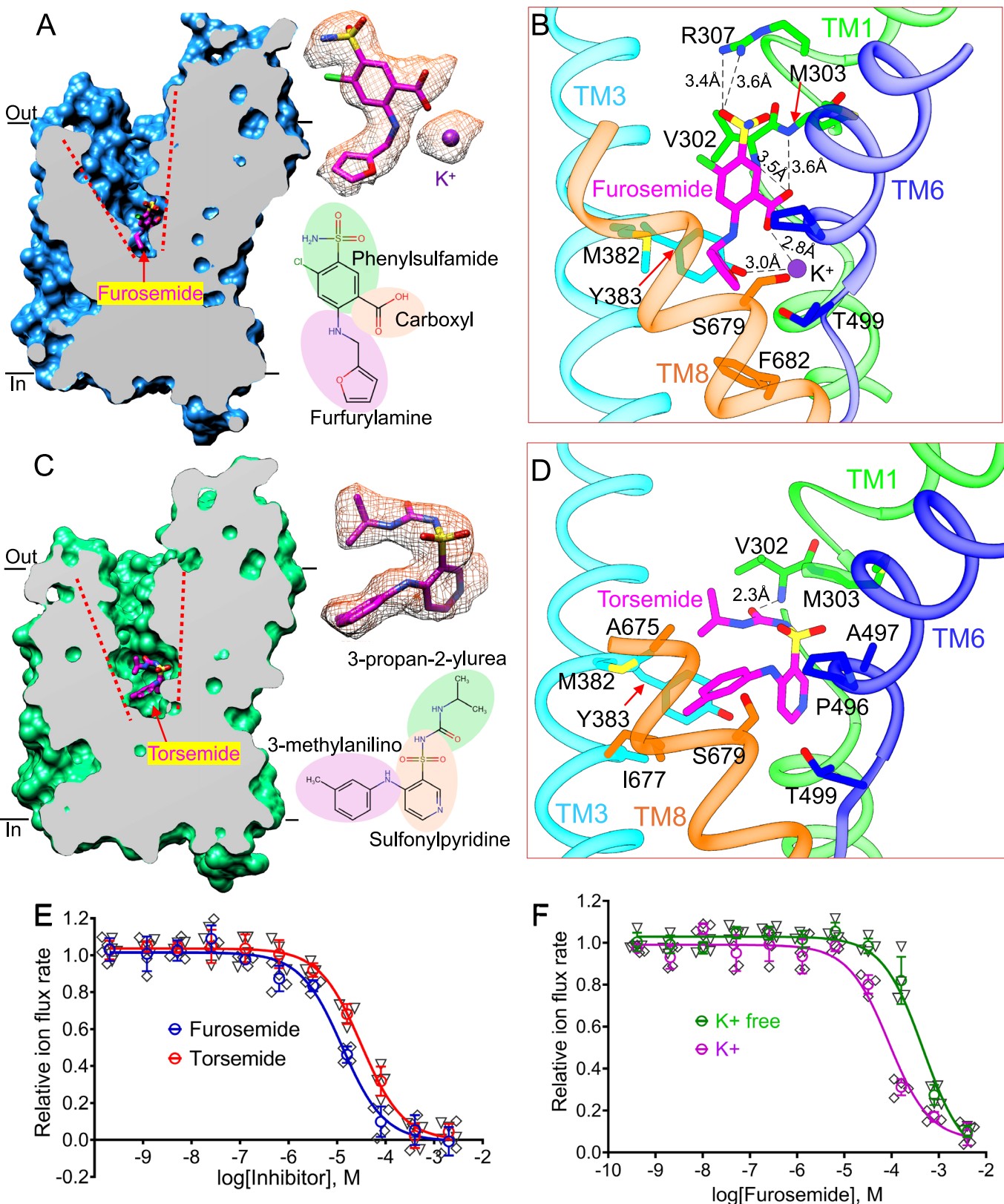

Taken together, our three pNKCC1/loop diuretics co-structures revealed two modes of NKCCs inhibition by loop diuretic drugs: the carboxyl group-bearing furosemide and bumetanide exhibit K+-dependent antagonism as their interactions with NKCCs require coordination and co-occlusion of a K+ ion, whereas torsemide lacks the carboxyl group and competes with K+ for binding to an overlapping site and its potency is likely attenuated by increasing [K+]. Brain-penetrable NKCC1 inhibitors have been sought to

**Figure 2. Furosemide and torsemide bind to an orthosteric site and occlude NKCC1 ion translocation path.**

(A) "Slab" view of the map highlights an extracellular vestibule obstructed by the furosemide in an outward-open state. Data information: Density with furosemide fitted inside and chemical structure of furosemide were shown. The density for furosemide was extracted from the original sharpened map in UCSF Chimera with contour level at 0.202. (B) A view of furosemide binding pocket highlights key coordinating residues. (C) "Slab" view of the map highlights an extracellular vestibule obstructed by the torsemide in an outward-open state. Data information: Density with torsemide fitted inside and chemical structure of torsemide were shown. The density for torsemide was extracted from the original sharpened map in UCSF Chimera with contour level at 0.204. (D) A view of torsemide binding pocket highlights key coordinating residues. (E) Dose–response curves of furosemide and torsemide were determined. Data information: Each square represents one kinetic measurement of a single sample supplied with furosemide or torsemide in designated concentration in a 96-well plate ($n = 4$ biological repeats; data are presented as mean values $+/-$ SD). The dose–response curves are fitted to the standard equation of log[Inhibitor] versus response (three parameters) using GraphPad Prism 8.0. (F) $IC_{50}$s of furosemide were determined in $K_2SO_4$ and $(NMDG)_2SO_4$ conditions. Data information: Each square represents one kinetic measurement of a single sample incubated with furosemide in a designated concentration in $K_2SO_4$ buffer or $(NMDG)_2SO_4$ buffer ($n = 4$ biological repeats; data are presented as mean values $+/-$ SD). The dose–response curves are fitted to the standard equation of log[furosemide] versus response (three parameters) using GraphPad Prism 8.0. Source data are available online for this figure.

restore the compromised inhibitory synapse transmission which is a hallmark of many psychiatric and neurodevelopmental diseases (Hampel et al, 2021; Welzel et al, 2023). Our NKCC1/torsemide co-structure should inspire the development of torsemide derivatives, which may cross the blood-brain barrier more readily than the charged bumetanide molecule, to inhibit NKCC1 in the central nervous system for the treatment of various brain disorders.

## ATP binds to an allosteric pocket in the CTD of NKCC1

In our NKCC1/bumetanide and NKCC1/torsemide maps, we observed some weak non-protein densities in an amphiphilic cavity in the CTD (Appendix Fig. S10). This cavity was recently assigned as a controversial loop diuretics binding site in NKCC1 (Flygaard et al, 2023; Moseng et al, 2023; Moseng et al, 2022), but an equivalent pocket was initially discovered as a receptor site for ATP in KCC1 and later in NCC as well (Chi et al, 2021; Fan et al, 2023; Zhao et al, 2024). To settle this debate, we prepared pNKCC1 sample supplemented with 100 μM ATP and 2 mM furosemide, yielding an NKCC1/furosemide map in which we clearly observed a non-protein density that matches the shape of an ATP molecule in this cavity of NKCC1 (Fig. 3A,B; Appendix Fig. S8). We also noticed an additional spherical density adjacent to the three phosphate groups of the ATP molecule which we tentatively assigned as a $Mg^{2+}$ ion because ATP is frequently complexed with $Mg^{2+}$ in cells (Buelens et al, 2021).

In our pNKCC1/furosemide structure, ATP-$Mg^{2+}$ binding stabilizes a loop (Gly925-Leu931) and a helix (Asn1002-Lys1021) that would otherwise be mobile and missing in all reported NKCC1 structures (Appendix Fig. S11); the region which encompasses amino acids 932–1001 and contains sorting signals for trafficking NKCC1 to the basolateral membrane in epithelia remains unresolved (Carmosino et al, 2008). When nestled within the pocket, ATP-$Mg^{2+}$ assumes a compact configuration such that its adenosine group packs against hydrophobic residues on one side of the pocket and its three phosphate groups, together with a coordinated $Mg^{2+}$ ion, interact with the polar residues on the other side (Fig. 3C). On the hydrophobic side, ATP's adenine group forms hydrogen bonds with the main chain oxygen atoms of Met794 and Val823, establishes hydrophobic contacts with His822, Val823, and Met794, and also forms cation-π interaction with Arg801. Substitution of Met794 with a bulky tryptophan (M794W) greatly decreased NKCC1 ion flux rate possibly due to sterically hindering ATP binding (Fig. 3D). ATP's ribose ring interacts with Arg801 via polar contact. On the other side dominated by

hydrophilic residues, ATP's three $PO_4^{3-}$ groups engage in extensive polar interactions with positively charged side chains of Arg801 and Lys889, as well as with the main chain nitrogen atoms of Lys890, Asp891, and Leu926. Meanwhile, the three phosphate groups additionally coordinate a $Mg^{2+}$ ion in a geometry that enables it to further participate in electrostatic interactions with the carboxyl group of Asp927. Of note, Arg801 appears to play a prominent role in anchoring ATP within the pocket as it simultaneously interacts with the adenine, the ribose, and the phosphate moieties of ATP. Indeed, weakening these interactions by R801A mutation significantly reduced NKCC1 ion transport activity (Fig. 3D). This key Arg801-$PO_4^{3-}$ interaction seen in NKCC1 is conserved in NCC (Arg654) and KCC1 (Lys699 and Lys707) (Figs. 3C and EV5), and mutation of Arg654 in NCC also diminishes its ion flux activity (Fan et al, 2023; Zhao et al, 2024). Other residues interact with ATP via their main chain nitrogen or oxygen atoms, and substitutions of these residues did not significantly impair NKCC1's ion flux activity (Appendix Fig. S12).

Overall, our structure agrees with the notion that a conserved ATP-binding site exists in the CTD of cation-chloride cotransporters (CCCs) (Chi et al, 2021; Fan et al, 2023; Zhao et al, 2024). ATP adopts similar poses in NCC, NKCC1, and KCC1 (Fig. EV5A), but the ATP-binding sites are not strictly conserved except for all these transporters harboring key basic residues to engage in polar interactions with ATP (i.e., Arg801 in NKCC1, Arg654 in NCC, and Lys699/Lys707 in KCC1) (Figs. 3C and EV5B–D). Variations in the ATP-binding pocket likely endow these CCCs with different binding affinity for ATP and, consequently, distinct responses to cellular ATP fluctuations. Indeed, we and others found that NCC remains complexed with endogenous ATP during purification (Fan et al, 2023; Zhao et al, 2024), whereas NKCC1 and KCC1 require supplemented ATP to reliably occupy the site in cryo-EM maps (Chi et al, 2021).

## NKCC1 phosphorylation strengthens coupling between its TMDs and CTDs

Our mass spectrometry data confirmed that NKCC1 harbors several key phosphoacceptor sites (Thr203, Thr205, Thr207, Thr212, Thr217, and Thr230) in an N-terminal segment of the transporter (Appendix Fig. S1) (Darman and Forbush, 2002; Schiapparelli et al, 2021), but it remains obscure how (de)phosphorylation of these residues, catalyzed by the opposing actions of WNKs-SPAK kinases and phosphatases, tunes its ion transport activity in response to physiological cues (e.g., cell

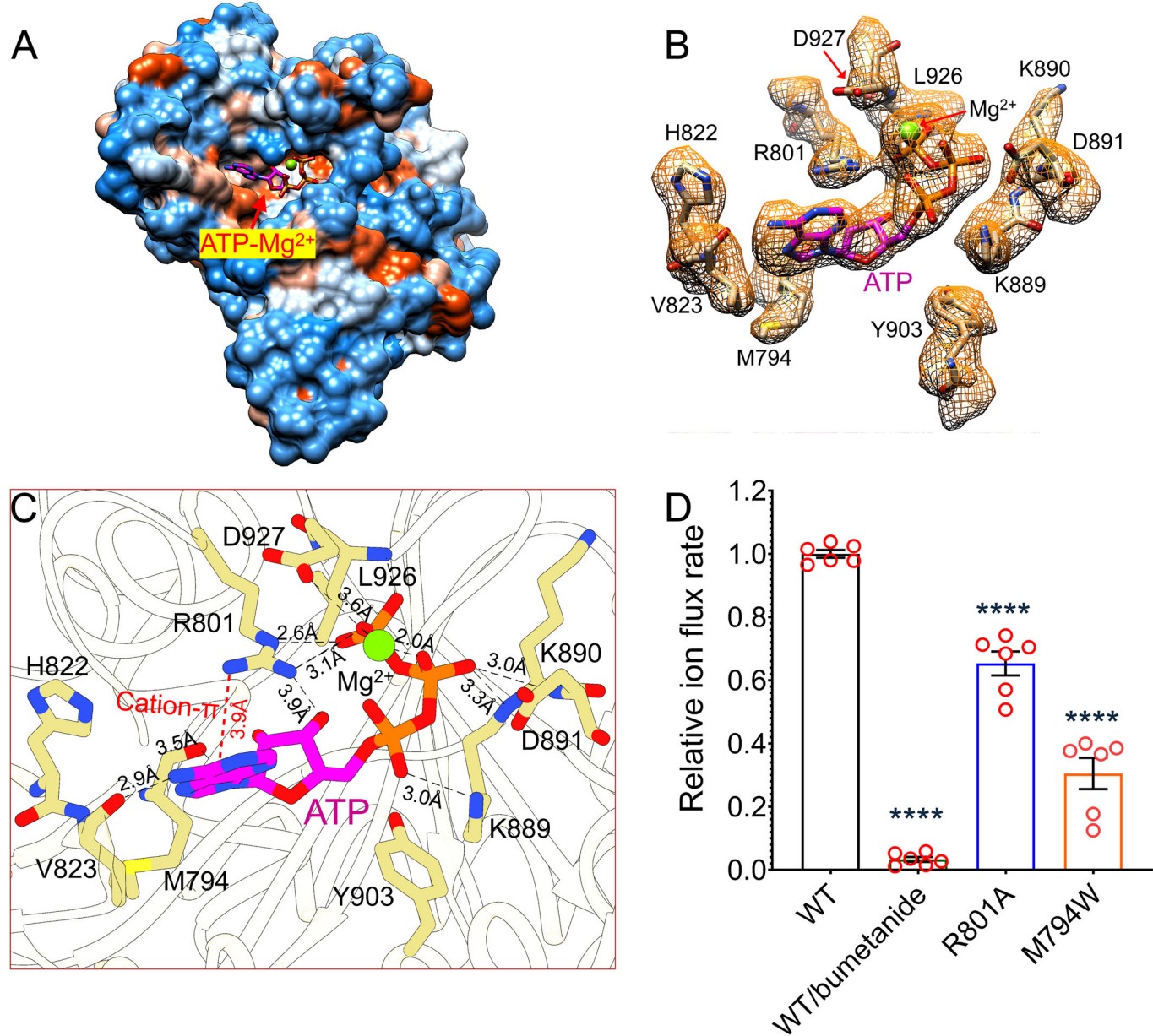

**Figure 3. ATP binds to an amphipathic pocket in CTD of NKCC1.**

(A) ATP nestles within an amphipathic pocket in CTD of NKCC1. Data information: The hydrophilicity of the CTD is displayed using the color code: dodger blue (hydrophilic), white (neutral), orange red (hydrophobic). (B) Well-resolved ATP-Mg²⁺ density enables modeling of the entire molecule. Data information: The densities for ATP-Mg²⁺ and its coordinating residues were extracted from original sharpened map and displayed at a contour level of 0.22 in UCSF Chimera. (C) An enlarged view highlights ATP-coordinating residues on NKCC1. (D) Perturbation of ATP-binding site reduced ion flux rate of NKCC1. Data information: Each circle represents one kinetic measurement of a single sample in a 96-well plate. Unpaired one-tailed Student's $t$ tests are used for statistical analyses ($n = 6$ biological repeats; data are presented as mean values $+/-$ SD). The $P$ values for WT versus WT/bumetanide, R801A, and M794W are 9.44E-14, 6.39E-05, 4.74E-06, respectively. ****$P < 0.0001$. Source data are available online for this figure.

shrinkage and swelling) (Alessi et al, 2014; Hoffmann et al, 2009). Our pNKCC1/furosemide map unambiguously defined the elusive N-terminal phosphoregulatory segment (residues Thr203-Leu250) which is seen to interact extensively with the swapped CTD of the second subunit and the intracellular loop 1 (ICL1; residues Thr343-Leu360) connecting the TM2 and TM3 helices (Fig. 4A,B). Within this N-terminal segment, two phosphoacceptors (i.e., Thr212 and Thr217) were clearly seen to each bear a negatively charged $PO_4^{3-}$

group; Thr230 resides in a flexible loop but could still be tentatively modeled with a $PO_4^{3-}$ group. These phosphate groups enable the three phosphoacceptors to interact with adjacent positively charged arginine and lysine residues on CTD and TMD (Fig. 4A,B). For instance, pThr212 and pT217 form salt bridges with the Lys1061/ Arg1064 and K1145/R1148 pair protruding from a concave surface on the swapped CTD, respectively. Thr203 forms a hydrogen bond with Lys1095 in the CTD. Our map was insufficient to resolve

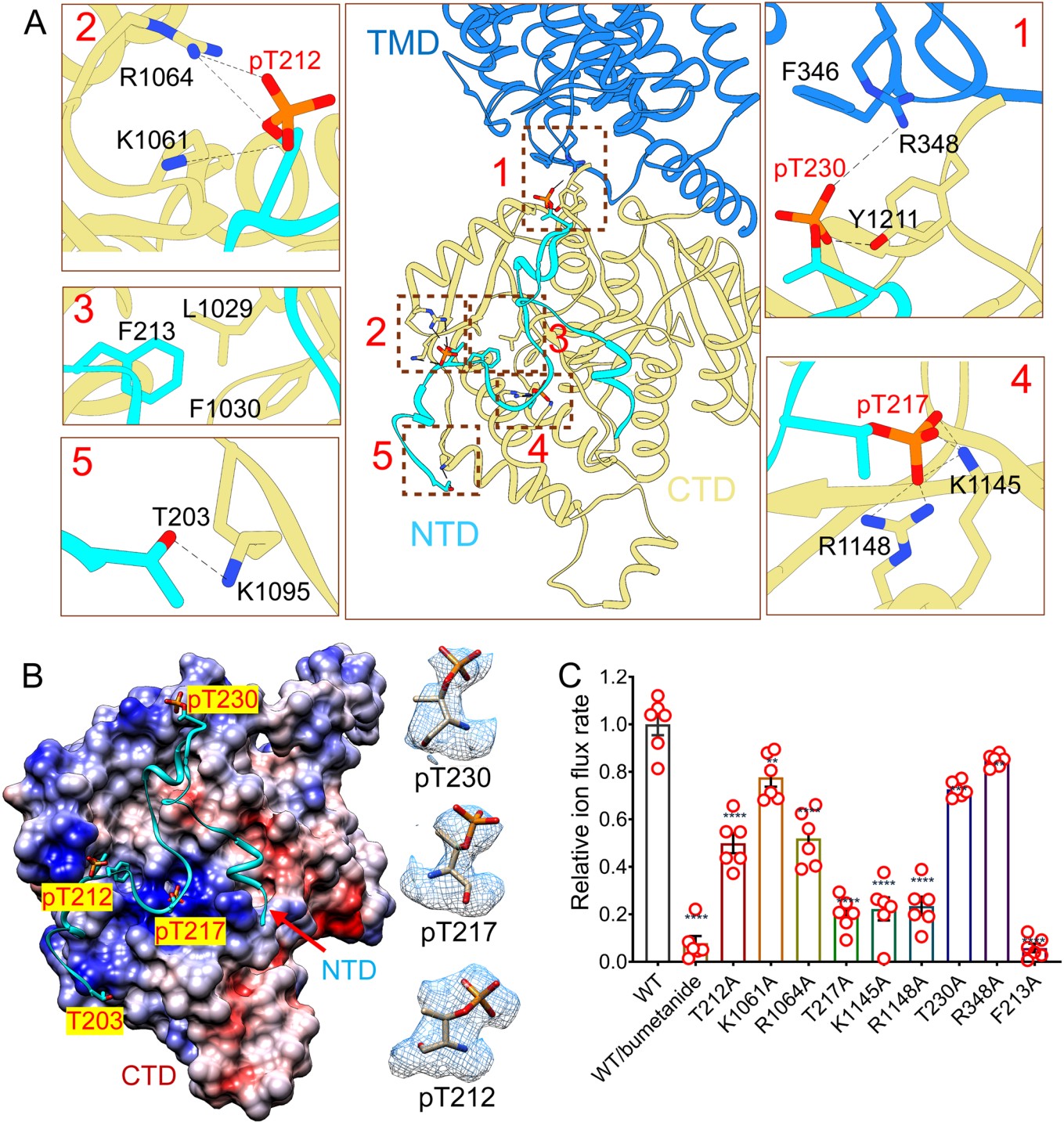

**Figure 4. Phosphorylation of an N-terminal segment promotes the communication between TMD and the cytosolic domain of NKCC1.**

(A) Phosphorylation of Thr212, Thr217, and Thr230 strengthens NTD-CTD and NTD-TMD association. (B) Phosphoacceptors foster polar interactions upon NKCC1 activation by kinases. Data information: pThr230, pThr217, and pThr212 are docked into corresponding density maps. The densities for pT212, pT217, and pT230 were extracted from original sharpened map and displayed at a contour level of 0.229, 0.229, 0.229 in UCSF Chimera, respectively. (C) Weakening phosphorylation-induced interactions abolishes or reduces NKCC1 ion transport rates. Data information: Ion flux rates were measured as the slopes of fluorescence change in the first 20 s. Each circle represents one kinetic measurement of a single sample in a 96-well plate. Unpaired one-tailed Student's $t$ tests are used for statistical analyses ($n = 6$ biological repeats; data are presented as mean values $+/-$ SD). The $P$ values for WT versus WT/bumetanide, T212A, K1061A, R1064A, T217A, K1145A, R1148A, T230A, R348A, and F213A are 2.09E-08, 5.72E-06, 2.10E-03, 1.22E-05, 2.13E-07, 1.95E-07, 8.43E-08, 6.15E-04, 9.08E-03, 6.66E-07, respectively. ****$P < 0.0001$; ***$P < 0.001$; **$P < 0.01$. All comparisons are WT versus mutants. Source data are available online for this figure.

phosphate at this site, but pThr203 could surely strengthen this polar interaction. Sitting just beneath the inner membrane, pT230 was seen to participate in electrostatic interactions with Arg348 in ICL1 and Tyr1211 in the extreme C-terminal tail. Besides these prominent $PO_4^{3-}$-mediated polar interactions, Phe213 apparently further stabilizes NTD/CTD association via hydrophobic interactions with Leu1029 and Phe1030. Structure-guided mutagenesis studies demonstrated that mutants designed to weaken these interactions (T212A, K1061A, R1064A, T217A, K1145A, R1148A, T230A, R348A, and F213A) all exhibit attenuated ion flux rate than the wildtype NKCC1 (Fig. 4C). Taken together, our structures and associated functional studies showed that Thr212, Thr217, and Thr230, once phosphorylated, serve as footholds for NTD to associate with CTD and TMD: pThr212 and pThr212 help to anchor NTD to a concave surface on CTD, while pThr230 facilitates the association of the resulting NTD/CTD structure with TMDs. These phosphorylation-dependent domain interfaces seen in our pNKCC1 structures may stabilize NKCC1 in a phospho-activated conformation that is conducive to rapid ion flux.

## ICL1 bridges cytosolic regulatory domains and ion transport path

In all three pNKCC1/loop diuretics co-structures, NKCC1 was captured in a similar dimeric architecture that features an extensive interface between the TMDs and cytosolic domains (Appendix Fig. S13), in contrast to most human NKCC1 structures where cytosolic domains were not resolved possibly because they are mobile with respect to the TMDs (Chew et al, 2019; Neumann et al, 2022; Yang et al, 2020; Zhang et al, 2021). The TMDs and cytosolic domains are coupled primarily via interactions among ICL1, intracellular loop 5 (ICL5), phospho-activated N-terminal segment, and the extreme C-terminal tail (Fig. 5A). Among them, the ICL1 was recognized as one of the most conserved structural motifs in all CCCs and it was hypothesized to gate ion exit from the intracellular side (Somasekharan et al, 2012; Zhao et al, 2022a; Zhao et al, 2022b). In our outward-open NKCC1 structures, ICL1, alongside other structural elements such as the intracellular gate formed by Lys624 and Asp510, indeed occludes an intracellular path for ion escape into the cytosol (Fig. 5B). On one side, the ICL1 is coupled to TM6b via a hydrogen bond formed between Gly350 (ICL1) and Ser508 (TM6b) (Fig. 5B). Disrupting this interaction by S508W mutation almost completely abolished NKCC1 ion transport activity (Fig. 5E). On the other side, Arg358 located at the short helix of ICL1 forms a salt bridge with Asp632 on TM8 (Fig. 5B); Arg358 additionally participates in cation-π interaction with Trp758 in a loop that connects TMDs and CTD (Fig. 5B). Abolishing these interactions by R358A, D632A, and W758A mutations all inactivated NKCC1 (Fig. 5E). The ICL1 and intracellular gate appear to be tightly coupled in NKCC1 because these structural motifs associate with each other and undergo a concerted movement as NKCC1 transitions between outward-open and inward-open conformations (Fig. 5C,D). Our structures hint that subtle displacement of ICL1 in response to (de)phosphorylation could propagate upward to affect the formation and rupture of the intracellular gate governed by Asp510 (TM6b) and Lys624 (TM8).

Besides engaging with the intracellular gate, the ICL1 also extensively associates with cytosolic domains beneath. In particular,

Arg348 (ICL1), pT230 (NTD), and Tyr1211 (the extreme C-terminal tail) form a triad that enables allosteric coupling and communications among these domains on which they reside (Fig. 5B). The fact that the phosphate group in pThr230 interacts with an arginine residue at the intracellular ion exit site strongly suggests that WNK/SPAK activates NKCC1 by strengthening association between TMDs and cytosolic domains. Although a mechanistic understanding of how ICL1 and regulatory phosphorylation of NTD act in concert to regulate NKCC1 ion translocation awaits future studies, it is conceivable that the phosphorylated N-terminal segment complexes with the CTD to enable isomerization among transport states via interaction with the ICL1.

## Discussion

Our structures of phosphorylated NKCC1 individually bound with the three most popular loop diuretic drugs (i.e., furosemide, torsemide, and bumetanide), combined with recent structural pharmacology studies of other CCCs such as KCC1/VU0463271 and NCC complexed with thiazide and thiazide-like diuretic drugs (Fan et al, 2023; Zhao and Cao, 2022; Zhao et al, 2024; Zhao et al, 2022b), unmistakably pinpointed an extracellular vestibule of CCCs ion translocation path as a hotspot targeted by distinct chemical types of orthosteric inhibitors. Our co-structures showed that loop diuretic drugs can readily access their receptor sites from the extracellular space, nicely explaining why these membrane-impermeable drugs can inhibit NKCCs within seconds. Overall, our three pNKCC1/loop diuretics co-structures cast further doubt that the reported intracellular bumetanide and furosemide binding site has any physiological bearing on their NKCCs antagonism (Moseng et al, 2022). Importantly, we discovered two modes of NKCCs antagonism by loop diuretics. Furosemide and bumetanide, and possibly other carboxyl group-bearing loop diuretics (e.g., piretanide), use the carboxyl group to coordinate and co-occlude a $K^+$ ion, and their potency is accordingly decreased when external $[K^+]$ drops. On the other hand, torsemide encroaches into and expels the $K^+$ ion from its binding site, suggesting that it could be more potent under low $[K^+]$ conditions. Given that normal urine $[K^+]$ is ~20 mM, furosemide and bumetanide may be less affected in inhibiting its therapeutic target NKCC2 than does torsemide. In the same vein, furosemide and bumetanide could be particularly superior in treating patients with elevated urine $[K^+]$ due to, for example, metabolic acidosis. Another intriguing finding regarding loop diuretic pharmacology is that the sulfamoyl group of furosemide and bumetanide directly engages with extracellular gate residues of NKCC1, but torsemide's sulfamoyl group reaches further toward the midway point of the ion translocation path and makes no obvious contact with NKCC1. It may thus be feasible to replace the sulfur atom of torsemide with, for example, a phosphorus atom, to enable sulfur-free loop diuretics. Such sulfur-free diuretics are urgently needed for the treatment of patients with sulfur allergies and are contraindicated to common sulfur-containing loop diuretics.

Our structures also showed in near-atomic details how ATP-$Mg^{2+}$ binds to an amphipathic pocket and stabilizes the CTD of NKCC1. An analogous ATP-binding site has been reported in KCC1 and NCC (Chi et al, 2021; Fan et al, 2023; Zhao et al, 2024), highlighting a potentially conserved role of ATP in regulating

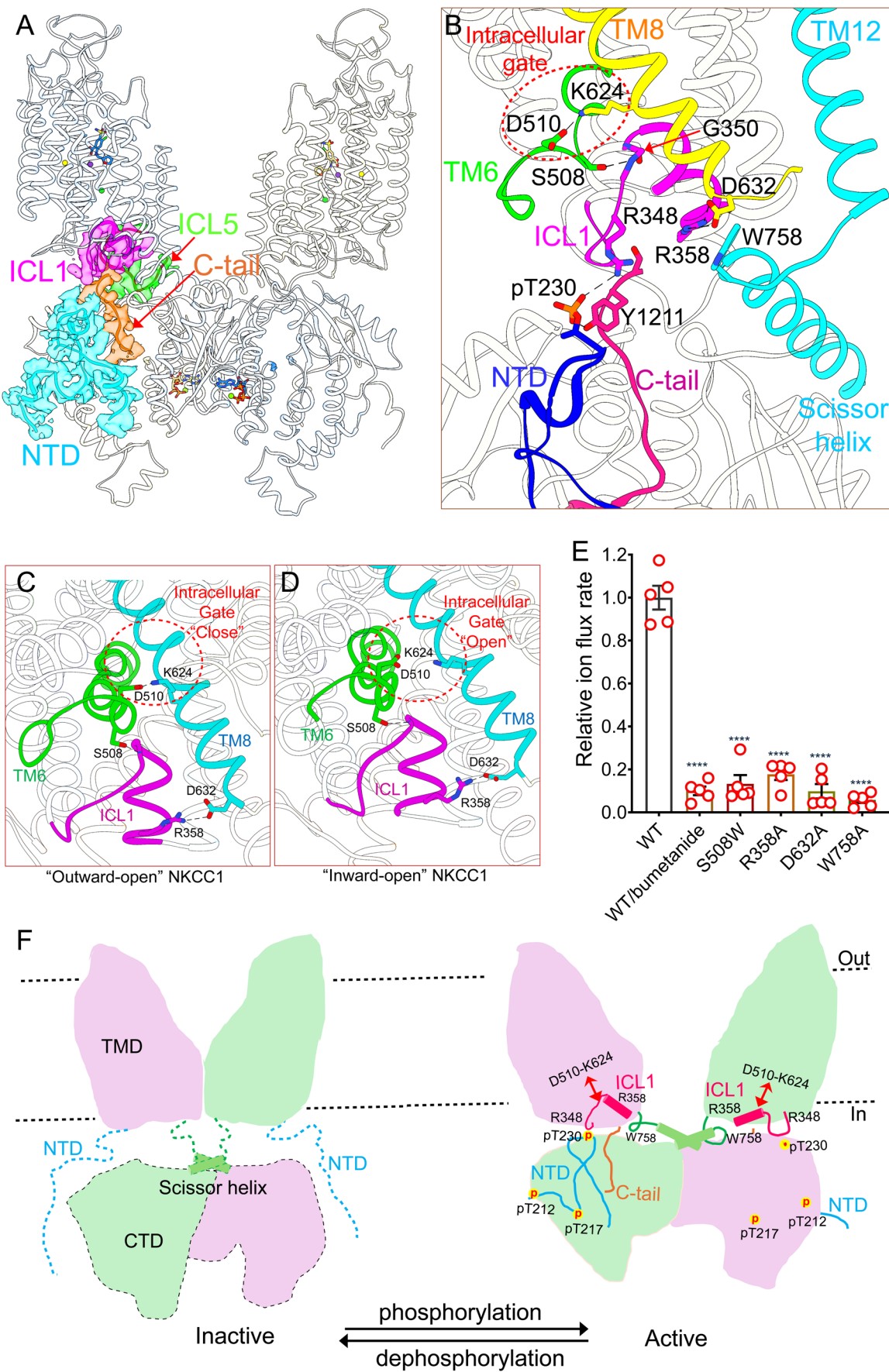

◀ **Figure 5.  The ICL1 plays a central role in regulating NKCC1.**

(**A**) The TMD-cytosolic domain interface primarily involves the phosphoregulatory N-terminal segment, C-tail, ICL1, and ICL5. (**B**) A zoomed view highlights associations centered around ICL1. (**C, D**) The allosteric associations between NKCC1 ICL1 and intracellular gate persist in both outward-open state (**C**) and in inward-open state (**D**). Note, the intracellular gate ruptures in the inward-open structure. (**E**) Disrupting the associations between the intracellular gate and ICL1 abolishes NKCC1 ion flux rate. Data information: Unpaired one-tailed Student's $t$ tests are used for statistical analyses ($n = 5$ biological repeats; data are presented as mean values $+/-$ SD). The $P$ values for WT versus WT/bumetanide, S508W, R358A, D632A, and W758A are 1.08E-05, 2.26E-06, 5.24E-06, 1.17E-06, 7.43E-06, respectively. ****$P < 0.0001$ for WT versus mutants. (**F**) A hypothetical model of NKCC1 regulation by (de)phosphorylation. In the dephosphorylated and inactive state (PDB code: 7ZGO), TMDs and cytosolic domains are disengaged, whereas two TMDs associate in the membrane. Phosphorylation of the NTD strengthens its association with the CTD, and together, interacts with the TMD, forming an allosteric network that couples pNTD, C-tail, and ICL1. Source data are available online for this figure.

CCCs. We showed that ATP-Mg$^{2+}$ binding stabilizes two dynamic C-terminal segments of NKCC1 which are otherwise unresolved in all previously reported NKCC1 structures (Chew et al, 2019; Moseng et al, 2022; Neumann et al, 2022; Yang et al, 2020; Zhang et al, 2021), hinting that ATP-Mg$^{2+}$ may act as a small "molecular glue" that allosterically stabilizes domain interfaces in NKCC1. A recent study showed that mutations in two mitochondrial transfer tRNA molecules underline a rare form of Gitelman's syndrome due to profoundly diminished NCC activity (Viering et al, 2022). It would be interesting to explore whether reduced mitochondrial ATP production in these patients can partly explain their NCC-related symptom and, if true, why NCC becomes so vulnerable as compared to many other proteins in the entire human proteome when cellular ATP level drops. We expect that a mechanistic understanding of the crucial allosteric role of ATP-Mg$^{2+}$ in CCCs will benefit from measuring how CCCs ion transport activity responds to ATP using liposome-based ion flux assay. It is also our hope that the ATP-Mg$^{2+}$ binding pocket of NKCCs could be targeted for the development of a new generation of diuretic drugs. Such allosteric modulators will also enrich CCCs pharmacology that has been historically dominated by the existing orthosteric loop and thiazide diuretics.

Our pNKCC1 structures strongly support a phosphoregulatory model whereby WNKs-SPAK phosphorylation promotes the association of NTD and CTD, which together go on to interact with and regulate the NKCC1 intracellular gate (Fig. 5F). We showed that three pThr residues are all tasked to strengthen intramolecular domain interfaces via PO$_4^{3-}$-mediated electrostatic interactions, and are thus incapable of simultaneously recruiting other cellular factors which could in turn regulate NKCC1 activity. Our studies also revealed that the ICL1 of NKCC1 plays an unexpected central role in transmitting conformational changes from the NTD phosphorylation sites to the intracellular gate. In this regard, NKCC2 intriguingly exists as three splicing isoforms that differ in the ICL1 (and TM2) amino acid sequences (Gimenez and Forbush, 2007; Gimenez et al, 2002). Future studies will determine whether these three NKCC2 isoforms respond differently to kinase activation and accordingly have distinct ion transport characteristics that are tailored to fulfill their specialized roles in distinct regions of the kidneys. A mechanistic understanding of NKCC1 phosphoregulation will also likely benefit from MD simulations and single-molecule fluorescence resonance energy transfer (FRET) studies, both of which can provide essential dynamic information currently missing in static cryo-EM structures. Nevertheless, our pNKCC1 structures should now enable rational targeting of the NKCCs phosphoregulatory interfaces to generate novel diuretic therapeutics for the treatment of edema and hypertension.

# Methods

### Reagents and tools table

| Reagent/resource | Reference or source | Identifier or catalog number |
|---|---|---|
| **Experimental models** | | |
| HEK293T/17 SF | ATCC | ACS-4500 (After initial purchase from ATCC, the cells were not authenticated of mycoplasma contamination in lab) |
| **Recombinant DNA** | | |
| Human NKCC1 | Gift from Dr. Forbush | Uniprot: P55011 |
| Human WNK1 | Gift from Dr. Rinehart | Uniprot: Q9H4A3 |
| Human SPAK | Synthesized by GENEWIZ | Uniprot: Q9UEW8 |
| Human Mo25 | Synthesized by GENEWIZ | Uniprot: Q9Y376 |
| mbYFPQS | Addgene | Cat#80742 |
| **Antibodies** | | |
| **Oligonucleotides and other sequence-based reagents** | | |
| **Chemicals, enzymes, and other reagents** | | |
| ATP | MilliporeSigma | Cat: A2383 |
| Bumetanide | MedChemExpress | Cat: HY-17468 |
| Furosemide | MedChemExpress | Cat: HY-B0135 |
| Torsemide | MedChemExpress | Cat: HY-B0247 |
| Biotin | IBA | Cat: 2-1016-002 |
| Strep-Tactin resin | IBA | Strep-tactin@XT 4flows |
| GDN | Anatrace | GDN-101 |
| LMNG-3 | Anatrace | NG310 |
| Calyculin-A | MedChemExpress | Cat: HY-18983 |
| HEK293 Freestyle medium | Invitrogen | Cat:12338002 |
| Cholesteryl hemisuccinate tris salt | Anatrace | CH210 |
| **Software** | | |
| Cryosparc | Punjani et al, 2017 | |
| RELION | Scheres, 2012; Zivanov et al, 2019 | |
| CTFFIND4 | Rohou and Grigorieff, 2015 | |

| Reagent/resource | Reference or source | Identifier or catalog number |
|---|---|---|
| MotionCorr | Zheng et al, 2017 | |
| TOPAZ | Bepler et al, 2019 | |
| 3DFSC | Tan et al, 2017 | |
| PHENIX1.18 | Adams et al, 2010 | |
| UCSF Chimera | Pettersen et al, 2004 | |
| Coot | https://www2.mrc-lmb.cam.ac.uk/personal/pemsley/coot/ Emsley and Cowtan, 2004 | |
| **Other** | | |
| Orbitrap Fusion Lumos mass spectrometer | Thermo Fisher | |
| nanoAcquity UPLC system | Water | |
| BioTek Synergy Neo2 HTS Multi-Mode Microplate Reader | Agilent Technologies | |

## Expression and purification of phospho-activated human NKCC1 protein

A full-length human NKCC1 construct (Uniprot: P55011) bearing an N-terminal Twin-Strep tag was cloned into a bi-directional PiggyBac transposon vector to generate HEK293T/17 SF (ATCC: ACS-4500; not authenticated of mycoplasma contamination in lab) stable cell lines for both structural and functional studies (Yusa et al, 2011). For structural studies, Twin-Strep-NKCC1 was cloned into the multiple cloning site I (MCSI) of the vector, and the WNK1 (residues 1-483 bearing S378D mutation)-Mo25-mbYFPQS-SPAK expression cassette was cloned into the MCSII of the vector; WNK1, Mo25, and SPAK are all human genes, and mbYFPQS is sensitive to [$Cl^-$] and bears an N-terminal myristoylation sequence for plasma membrane targeting (Watts et al, 2012), and these four genes are linked by a P2A sequence (Kim et al, 2011). For determining the NKCC1/furosemide co-structure, we used human NKCC1 bearing K289N and A492E mutations which was shown to exhibit enhanced affinity for loop diuretics (Dehaye et al, 2003; Zhao et al, 2022a). For functional studies, Twin-Strep-NKCC1 and mutants were cloned into the MCSI site and mbYFPQS was cloned into the MCSII site.

To prepare pNKCC1/bumetanide sample, the stable cell line co-expressing NKCC1 and kinases was grown in suspension in Freestyle 293 expression medium (Invitrogen, Carlsbad, CA) at 37 °C in an orbital shaker; protein expression was induced by adding 1 µg/ml doxycycline, together with 5 mM sodium butyrate to boost transcription, when the cell density reached ~1.5 × 10$^6$/ml. 20 µM bumetanide was added during protein expression to suppress cell death caused by constitutive WNK1 and NKCC1 activation. Cells were harvested 36 h post induction, and then suspended into a hypotonic, $Cl^-$ free buffer composed of 20 mM HEPES (pH 7.4), 45 mM (NMDG)$_2$SO$_4$, supplemented with 100 µM bumetanide for another 2 h with gentle shaking at 37 °C. Cells were then broken with a Dounce homogenizer, and

membranes were harvested by centrifugation at 40,000 rpm for 1 h and resuspended in a buffer composed of 20 mM HEPES (pH 7.4), 50 mM Na$_2$SO$_4$, 25 mM K$_2$SO$_4$, 5 mM KCl, 100 µM bumetanide, and flash-frozen in liquid nitrogen and stored at −80 °C until use.

All protein purification steps were carried out at 4 °C unless stated otherwise. Membrane proteins were extracted for 2 h at 4 °C in a buffer composed of 20 mM HEPES (pH 7.4), 50 mM Na$_2$SO$_4$, 25 mM K$_2$SO$_4$, 5 mM KCl, 100 µM bumetanide, 3 mM lauryl maltose neopentyl glycol (LMNG-3), and 0.6 mM cholesteryl hemisuccinate tris salt (CHS), 5 µg/ml leupeptin, 1.4 µg/ml pepstatin A, and 2 µg/ml aprotinin. The supernatant was collected after centrifugation at 18,000 rpm for 30 min and then incubated with Strep-Tactin resin (IBA, Strep-tactin@XT 4flows) for 2 h. Contaminant proteins were removed by washing with 15 column volume of buffer composed of 20 mM HEPES (pH 7.4), 50 mM Na$_2$SO$_4$, 25 mM K$_2$SO$_4$, 5 mM KCl, 100 µM bumetanide, and 0.02% GDN. NKCC1 protein was eluted from the resin with a buffer composed of 20 mM HEPES (pH 7.4), 50 mM Na$_2$SO$_4$, 25 mM K$_2$SO$_4$, 5 mM KCl, 100 µM bumetanide, 50 mM biotin, and 0.02% GDN. NKCC1 was further separated with a Superose 6 column using a buffer composed of 20 mM HEPES (pH 7.4), 50 mM Na$_2$SO$_4$, 25 mM K$_2$SO$_4$, 5 mM KCl, 100 µM bumetanide, and 0.01% GDN; peak fractions corresponding to NKCC1 were pooled and then concentrated for cryo-EM analyses.

The pNKCC1/torsemide sample was similarly prepared as pNKCC1/bumetanide except for excluding K$^+$ in the purification buffer composed of 20 mM HEPES (pH 7.4), 75 mM Na$_2$SO$_4$, 100 µM torsemide, and 0.02% GDN. To prepare the pNKCC1/furosemide sample, 0.5 µM calyculin-A (a phosphatase inhibitor) was included during the hypotonic and $Cl^-$ free buffer treatment for enhancing phosphorylation, and 100 µM ATP was added to all the buffers during purification. As furosemide is of low potency, 2 mM furosemide was used to complex with pNKCC1 for cryo-EM analyses.

## $Cl^-$ influx assay in HEK293 cells

The NKCC-mediated $Cl^-$ influx was measured using a membrane-targeted yellow fluorescent protein (mbYFPQS) as a $Cl^-$ indicator (Watts et al, 2012). Briefly, ~1.0 × 10$^5$ Twin-Strep-NCC/mbYFPQS stable HEK293 cells were seeded per well in a poly-D-lysine treated, black-walled, clear-bottom 96-well plate, and 1 µg/ml doxycycline was added to induce protein expression. HEK293 cells only expressing mbYFPQS was included as a negative control in all experiments. In all, 24–36 h post induction, the medium was replaced by 100 µl activation buffer (20 mM HEPES, 70 mM (NMDG)$_2$SO$_4$, pH 7.4), and incubated for 2–3 h prior to assay. The activation buffer was exchanged to 100 µl assay buffer (20 mM HEPES, 90 mM NaCl, 50 mM KCl, pH 7.4) to initiate NKCC-mediated $Cl^-$ influx; 100 µM loop diuretics drugs was also added in both activation and assay buffer to validate that observed $Cl^-$ influx was mediated by NKCC1. Fluorescence intensity was measured on a BioTek Synergy Neo2 HTS Multi-Mode Microplate Reader (excitation/emission wavelengths are 485 nm/535 nm). The rates of $Cl^-$ transport were calculated as the slopes of the fluorescent intensity change within the initial 20 s. Unpaired Student's *t* test was used to evaluate the significance of NKCC1-mediated, loop diuretics-sensitive transport activity.

For the assay to measure the effect of external $K^+$ on $IC_{50}$s of furosemide, the $K^+$-containing buffer (20 mM HEPES, 70 mM $K_2SO_4$, pH 7.4) and the $K^+$-free buffer (20 mM HEPES, 70 mM $(NMDG)_2SO_4$, pH 7.4) was used to activate NKCC1; furosemide at a series of concentrations were added to the activation buffers. As the $K^+$ is an obligatory substrate ion for $Cl^-$ influx, the assay buffer (20 mM HEPES, 90 mM NaCl, 50 mM KCl, pH 7.4) without furosemide was used to initial the assay. The binding affinities of loop diuretics in $K^+$-containing and $K^+$-free conditions during the activation step were measured by fitting with dose–response curves.

## Phosphorylation analysis

To identified phosphorylation sites, NKCC1 proteins were processed by using a procedure described in our recent publications (Wu et al, 2024; Wu et al, 2022). In brief, proteins were reduced with dithiothreitol (DTT) and alkylated with iodoacetamide followed by digestion on a S-Trap column (ProtiFi, LLC) by using either trypsin or chymotrypsin. The resulting peptides were analyzed by an Orbitrap Fusion Lumos mass spectrometer (Thermo Fisher) integrated with a nanoAcquity UPLC system (Waters) system. A 90-min gradient of buffer A (2% ACN, 0.1% formic acid) and buffer B (0.1% formic acid in ACN) was used for separation: 1% buffer B at 0 min, 5% buffer B at 1 min, 22% buffer B at 60 min, 36% buffer B at 75 min, 50% buffer B at 80 min, 90% buffer B at 85 min, 90% buffer B at 90 min. All the MS data were acquired in data-dependent acquisition mode, with parameters described previously. Database searching of the raw files was performed in Proteome Discoverer 2.4 (Thermo Fisher Scientific) with the Sequest HT search engine by using the customized database of human NKCC1. The database-searching parameters of phosphopeptides were set as below: full tryptic digestion and allowed up to two missed cleavages (for peptides derived from trypsin digestion) and full chymotryptic digestion and allowed up to four missed cleavages (for peptides derived from chymotrypsin digestion), the precursor mass tolerance was set at 10 ppm, whereas the fragment mass tolerance was set at 0.02 Da. Carbamidomethylation of cysteines ( + 57.0215 Da) was set as a fixed modification, and variable modifications of methionine oxidation (+ 15.9949 Da), acetyl (N-terminus, +42.011 Da), and phosphorylation (serine (S), threonine (T) or tyrosine (Y), +79.966 Da) were allowed. The false-discovery rate (FDR) was determined by using a target-decoy search strategy. The phosphosites with greater than 0.75 localization probabilities were considered as confident identification.

## Electron microscopy sample preparation and data collection

For cryo-EM, 5 µl of NKCC1 sample at ~3–5 mg/ml was applied to a glow-discharged Au 1.2/1.3 holey, 300 mesh gold grid and blotted for 2.5 s at 4 °C, 90% relative humidity on a Vitrobot Mark III (FEI) before being plunge-frozen in liquid ethane cooled by liquid nitrogen. Data were collected on a Krios (FEI) operating at 300 kV equipped with the K3 direct electron detector and a GIF energy filter at the University of Utah, National Center for Cryo-EM Access and Training (NCCAT), SLAC, NCI National Cryo-EM facility, and Pacific Northwest Cryo-EM Center (PNCC). Movies

were recorded using SerialEM or EPU, with a defocus range between −1.0 and −3.5 µm. Specifically, movies were recorded in super-resolution counting mode at a physical pixel size of 0.8256 Å (NKCC1/furosemide), 0.83 Å (NKCC1/torsemide), and 1.06 Å (NKCC1/bumetanide). The data were collected at a dose rate of 1.0–1.25 e − /$Å^2$/frame with a total exposure of 40–50 frames, giving a total dose of 40–50 e − /$Å^2$.

## Image processing 3D reconstruction and model building

Movie frames were aligned, dose-weighted, and then summed into a single micrograph using MotionCor2 (Zheng et al, 2017). CTF parameters for micrographs were determined using the program CTFFIND4 (Rohou and Grigorieff, 2015). Approximately 2000 particles were manually boxed out in relion4 to train a neuronal network model which was then used to extract particles from all micrographs using TOPAZ (Bepler et al, 2019). For the NKCC1/furosemide dataset, a total of 8,051,818 particles were extracted and then subjected to one round of 2D classification in cryoSPARC 3.0 software (Punjani et al, 2017). "Junk" particles that were sorted into incoherent or poorly resolved classes were rejected from downstream analyses. The remaining 1,135,556 particles from well-resolved 2D classes were pooled and were used to calculate four ab initio models in cryoSPARC 3.0 without imposing any symmetry followed by heterogenous refinement. The particles from the good class were subjected CTF correction and polishing in RELION4 software (Scheres, 2012; Zivanov et al, 2019) and then used to calculate a 2.7 Å map using non-uniform refinement in cryoSPARC 3.0.

For the NKCC1/torsemide dataset, a total of 2,360,417 particles were extracted and subjected to one round of 2D classification in cryoSPARC 3.0, resulting in 708,197 good particles. Ab initio models and heterogeneous refinement were similarly carried out in cryoSPARC 3.0 as described for the NKCC1/furosemide dataset. The good class consisting of 67,584 particles were subjected to CTF refinement and Bayesian polishing in RELION4 (Zivanov et al, 2019) and yielded a final map of 2.6 Å resolution after non-uniform refinement in cryosparc 3.0.

For the NKCC1/bumetanide dataset, a total of 1,694,969 particles were extracted and subjected to one round of 2D classification in cryoSPARC 3.0, resulting in 563,098 good particles. Ab initio models and heterogeneous refinement were similarly carried out in cryoSPARC 3.0 as described for the NKCC1/ furosemide dataset. The good class consisting of 90,380 particles were subjected to CTF refinement and Bayesian polishing in relion4 and yielded a final map of 2.5 Å resolution after non-uniform refinement in cryosparc 3.0. To evaluate potential orientation bias, directional resolutions of the three maps were calculated using a remote 3DFSC processing server (Tan et al, 2017).

The three maps were initially sharpened in cryoSPARC 3.0 with an overall b factor of -88 $Å^2$ (pNKCC1/furosemide), −63 $Å^2$ (pNKCC1/torsemide), and −67 $Å^2$ (pNKCC1/bumetanide) for model building in Coot 0.8.9.3 software. In the later stages of model building, the maps were locally sharpened in PHENIX1.18 (Adams et al, 2010) to further enhance their clarity. All cif files for ligands (i.e., ATP, bumetanide, torsemide, and furosemide) with geometric constraints were generated in Coot using the functionality of ligand builder. The models were refined in real space against the locally sharpened maps using PHENIX1.18 software with

several iterations; in earlier iterations, morphing and simulated annealing were carried out to bring some poorly fitted segments of the models into the densities, and in the last iteration only default geometric constraints were applied. Final models were assessed in MolProbity as shown in Appendix Table S1. UCSF Chimera was used to visualize and segment density maps, and to generate figures (Pettersen et al, 2004).

### Molecular dynamics simulations

Model systems were constructed similar to previous studies (Ruiz Munevar et al, 2024; Zhao et al, 2022a). Human NKCC1 monomer encompassing the transmembrane region (residues 282 to 753) was embedded in a POPC bilayer. The polypeptide encompasses the acetyl and N-methyl groups at the N- and C-termini, respectively. Missing side chain atoms were added and refined with MODELLER (Webb and Sali, 2016). The ionization state of titratable residues was determined using PropKa 3.0 (Olsson et al, 2011) and assuming pH 7.0. The presence of water molecules in buried protein cavities was assessed with DOWSER (Zhang and Hermans, 1996). Two forms of neutral torsemide were considered, namely: (i) the protonated pyridine and deprotonated sulfonylurea (Tor_NH_N, Fig. EV4D) and (ii) the neutral pyridine and neutral sulfonylurea (Tor_N_NH, Fig. EV4D). Furosemide features a carboxylic group which is predicted to be predominantly deprotonated at pH 7.0. The model included hNKCC1 and bound torsemide, surrounded by 306 POPC molecules, 60 $Cl^-$, 30 $K^+$, 30 $Na^+$ ions (including the experimentally determined $Cl^-$ and $Na^+$ ions) and ~23000 water molecules, for a total of ~116,000 atoms in a simulation box of ~$95 \times 110 \times 107$ $Å^3$ size.

MD simulations were performed with the GPU version of the PMEMD code of the AMBER package (Dickson et al, 2014). The system was treated under periodic boundary conditions, using the particle mesh Ewald method to compute long-range electrostatics (Darden et al, 1993). A 10 Å cutoff was used for the real part of the electrostatic and for van der Waals interactions. An integration time step of 2 fs was used, while constraining bonds involving hydrogen atoms (Ryckaert et al, 1977). Simulations were performed at constant temperature (300 K) and pressure (1 bar). The lipid bilayer, water solvent, and counterions were equilibrated around the protein during 200 ns MD simulation. After energy minimization, the system was gradually heated to 300 K, restraining protein backbone atoms to the experimental structure. For each protonation state of torsemide, three replicas were simulated starting from different initial conditions. For each replica, 250 ns production MD were performed.

### Conformational analysis

Quantum chemical calculations were performed with the software ORCA (Neese, 2012) to determine the relative stability of different conformations of the isolated Tor_NH-N molecule. Density functional calculations were performed using the hybrid B3LYP (Becke, 1993) functional and basis set 6-31 G(d) (Hariharan and Pople, 1973).

## Data availability

The cryo-EM maps have been deposited in the Electron Microscopy Data Bank (EMDB) with the accession codes EMD-45081

(pNKCC1/furosemide), EMD-45083 (pNKCC1/torsemide), and EMD-45084 (pNKCC1/bumetanide). The atomic coordinates for the corresponding maps have been deposited in the Protein Data Bank (PDB) with the accession 9C0E (pNKCC1/furosemide), 9C0G (pNKCC1/torsemide), and 9C0H (pNKCC1/bumetanide). All other data and reagents that support the findings of this study are available from the corresponding author upon request with no restrictions.

The source data of this paper are collected in the following database record: biostudies:S-SCDT-10_1038-S44318-025-00368-6.

## Peer review information

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

## Acknowledgements

This work was supported by the NIH grant R01 DK128592 to E.C. and the NIH grant GM117230 to J.R. We thank Anita Orendt, Irvin Allen, Martin Cuma, and other staff members at the Utah Center for High-Performance Computing for computational support. We are grateful to Barbie Ganser and David Belnap for data collection at the Electron Microscope Core at the University of Utah. We thank Omar Duvulcu, Ed Eng, Elina Kopylov, Grace Nye, Partrick Mitchell, and other staff members at the PNCC, S²C², and NCCAT for data collection and technical support. The Electron Microscope Core at the University of Utah was supported by a grant from the Beckman Foundation. A portion of this research

was supported by NIH grant U24GM129547 and performed at the PNCC at OHSU and accessed through EMSL (grid.436923.9), a DOE Office of Science User Facility sponsored by the Office of Biological and Environmental Research. Some of this work was performed at the National Center for CryoEM Access and Training (NCCAT) and the Simons Electron Microscopy Center located at the New York Structural Biology Center, supported by the NIH Common Fund Transformative High-Resolution Cryo-Electron Microscopy program (U24 GM129539,) and by grants from the Simons Foundation (SF349247) and NY State Assembly. Some of this work was performed at the Stanford-SLAC Cryo-EM Center ($S^2C^2$), which is supported by the National Institute of General Medical Sciences (1R24GM154186). The content is solely the responsibility of the authors and does not necessarily represent the official views of the National Institutes of Health.

## Author contributions

**Yongxiang Zhao**: Conceptualization; Investigation; Writing—original draft; Writing—review and editing. **Pietro Vidossich**: Data curation. **Biff Forbush**: Conceptualization. **Junfeng Ma**: Data curation. **Jesse Rinehart**: Conceptualization. **Marco De Vivo**: Data curation. **Erhu Cao**: Conceptualization; Supervision; Funding acquisition; Writing—review and editing.

Source data underlying figure panels in this paper may have individual authorship assigned. Where available, figure panel/source data authorship is listed in the following database record: biostudies:S-SCDT-10_1038-S44318-025-00368-6.

## Disclosure and competing interests statement

The authors declare no competing interests.

# Expanded View Figures

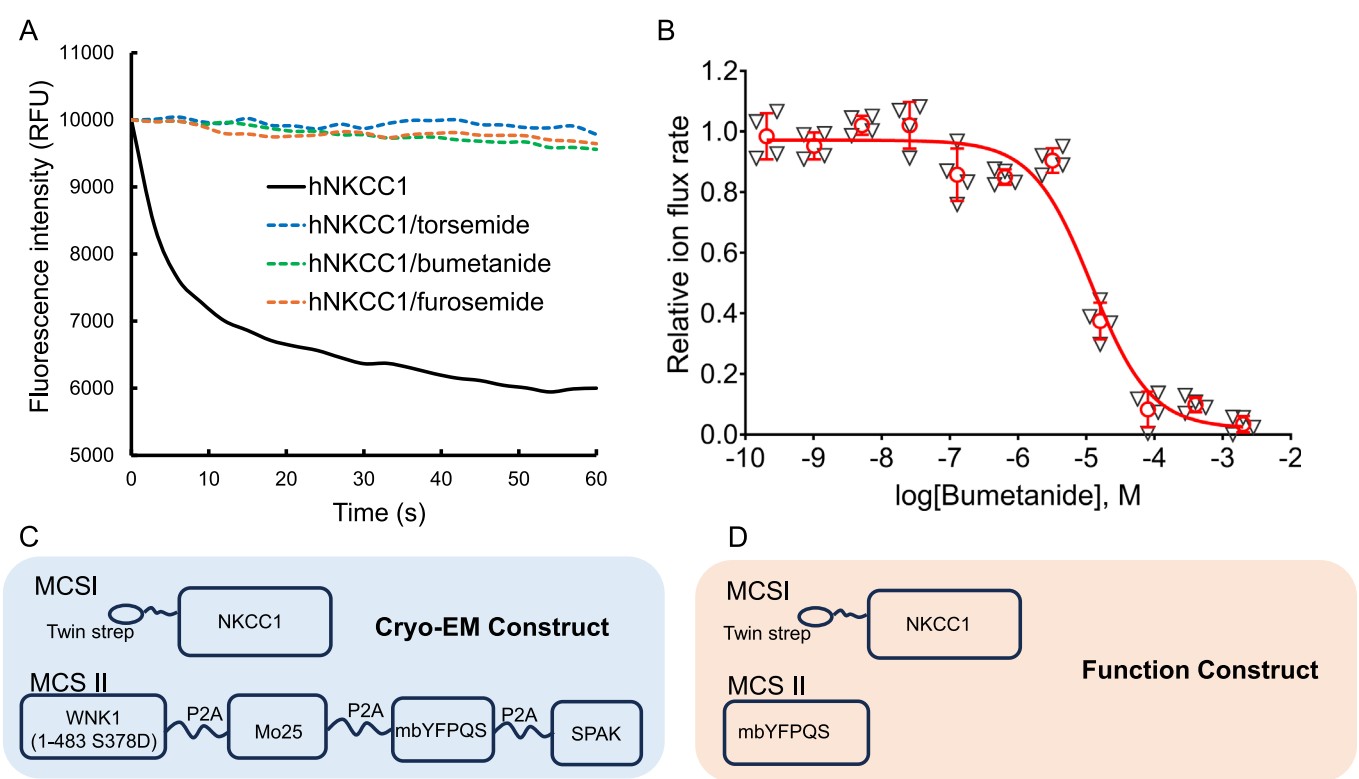

Figure EV1. **Functional characterization of human NKCC1 in HEK293 cells.**

(**A**) NKCC1-mediated Cl⁻ influx was inhibited by furosemide, bumetanide, and torsemide applied externally to the cells at 100 µM concentration. (**B**) Dose–response curve was determined for NKCC1 inhibition by bumetanide. Data information: Each triangle represents one kinetic measurement of a single sample incubated with bumetanide of indicated concentration in a 96-well plate ($n = 4$ biological repeats; data are presented as mean values $+/-$ SD). The dose–response curve is fitted to the standard equation of log[bumetanide] versus response (three parameters) using GraphPad Prism 8.0. (**C**) Design of NKCC1 constructs for cryo-EM studies. (**D**) Design of NKCC1 constructs for functional studies.

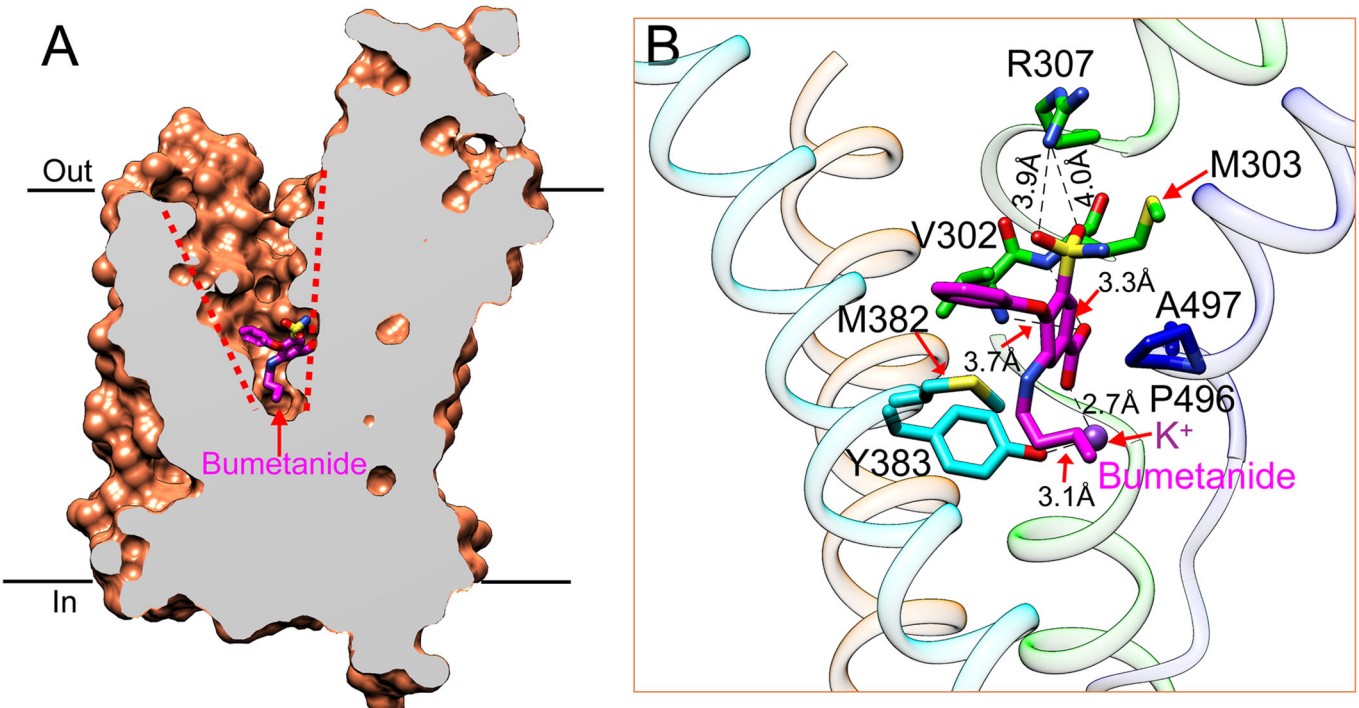

**Figure EV2.    Bumetanide binds to the NKCC1 extracellular vestibule and traps it in an outward-open state.**

(**A**) A "cut-off" view of pNKCC1/bumetanide co-structure highlights an extracellular vestibule in which bumetanide resides. (**B**) A view of bumetanide binding pocket highlights the key coordinating residues.

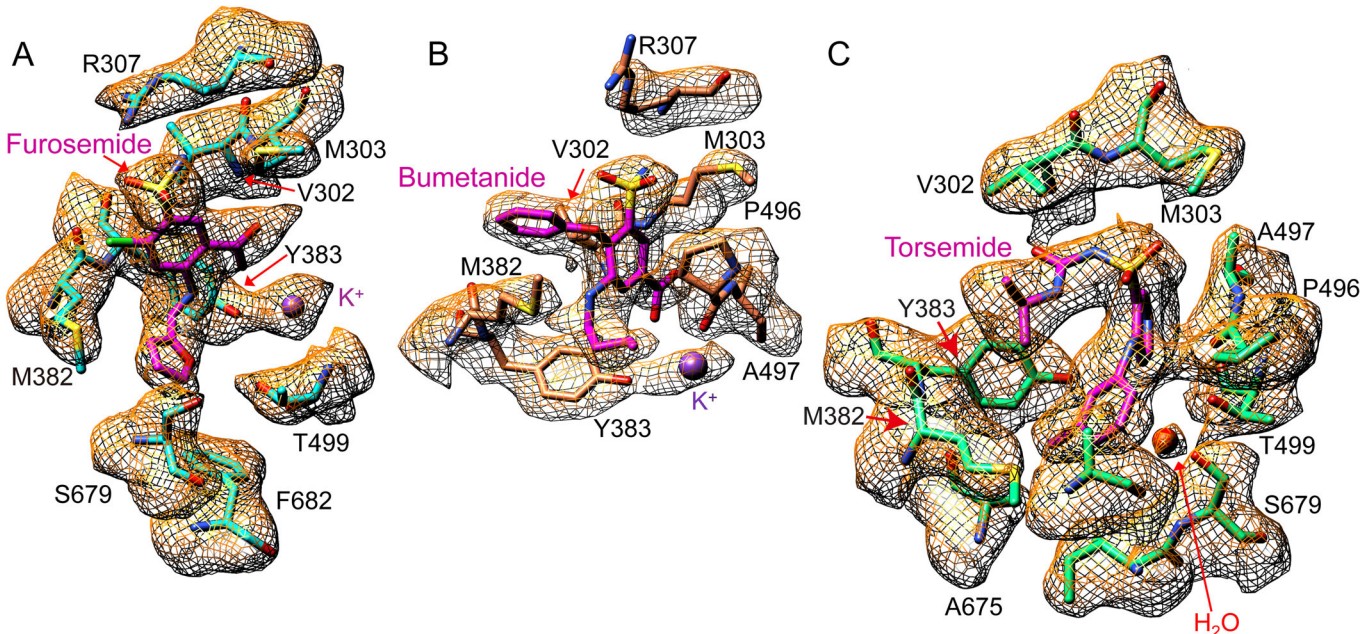

**Figure EV3. Chemical environment of loop diuretics binding sites.**

(A) Well-resolved furosemide, K$^+$, and their coordinating residues were docked into cryo-EM densities. Data information: The densities for furosemide, K$^+$, and its coordinating residues were extracted from original sharpened map and displayed at a contour level of 0.202 in UCSF Chimera. (B) Well-resolved bumetanide, K$^+$, and their coordinating residues were docked into cryo-EM densities. Data information: The densities for bumetanide, K$^+$, and its coordinating residues were extracted from original sharpened map and displayed at a contour level of 0.241 in UCSF Chimera. (C) Well-resolved torsemide, water, and their coordinating residues were docked into cryo-EM densities. Data information: The densities for torsemide, water, and their coordinating residues were extracted from original sharpened map and displayed at a contour level of 0.141in UCSF Chimera.

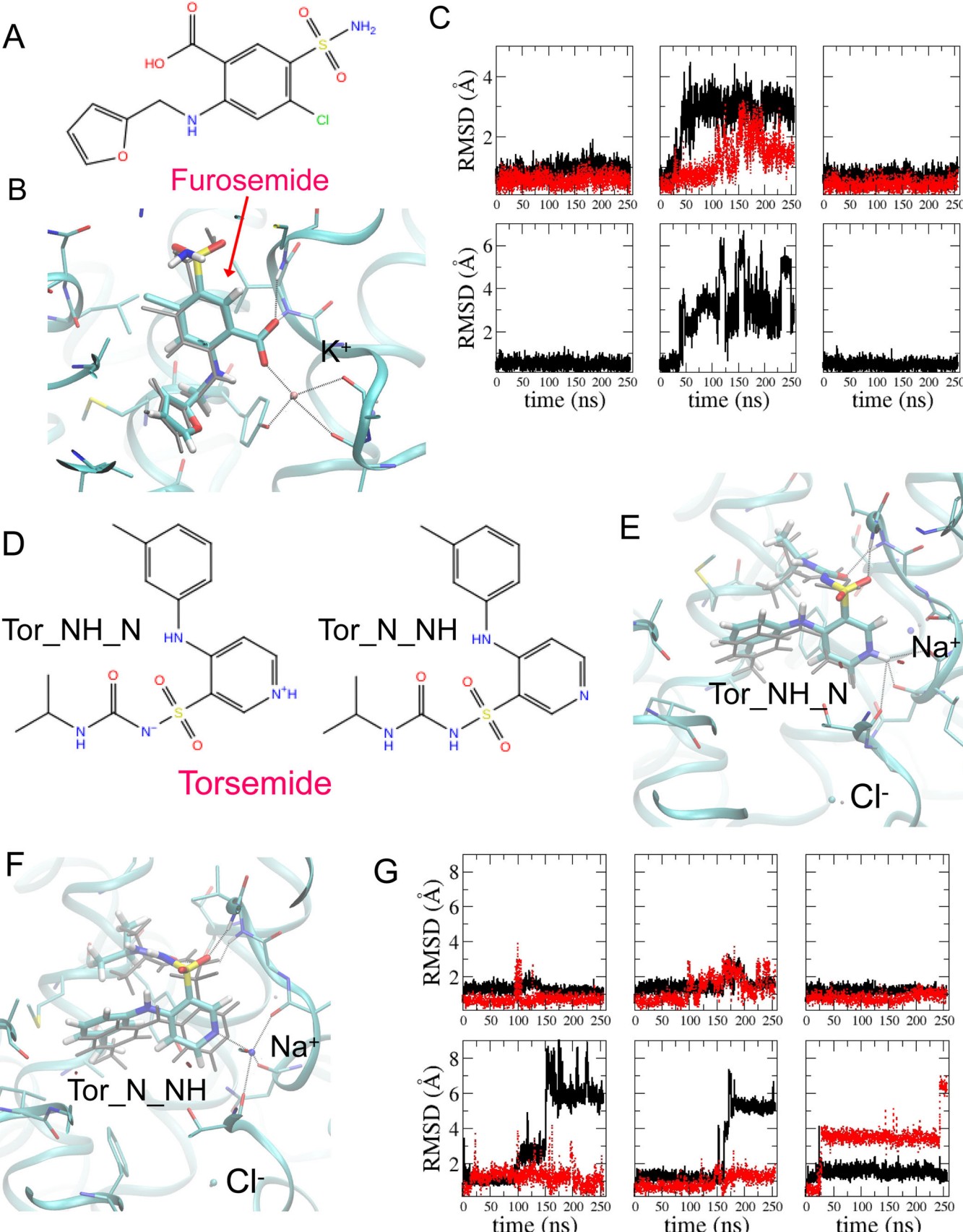

◀ **Figure EV4. Molecular dynamics simulations of furosemide and torsemide when bound to NKCC1.**

(A) Chemical structure of furosemide. (B) Structure after 250 ns simulation of furosemide; the initial experimental pose of furosemide and $K^+$ are shown in gray. (C) Root mean square displacement (RMSD) with respect to the initial pose of furosemide (black line) and $K^+$ ion (red dotted line) showed in the top three panels and of $Na^+$ showed in the bottom three panels. Three replicas of the NKCC1/furosemide complex were simulated. Note, in the second simulation (the two middle panels), the high mobility of $Na^+$ likely also causes instability of furosemide in the pocket. (D) Chemical structures of two possible forms of torsemide in neutral pH are shown. (E) Structure after 250 ns simulation of Tor_NH_N; the initial pose of torsemide, $Na^+$ and $Cl^-$ are shown in gray. (F) Structure after 250 ns simulation of Tor_N_NH; the initial pose of torsemide, $Na^+$ and $Cl^-$ are shown in gray. (G) RMSD with respect to the initial pose of torsemide (black line) and $Na^+$ ion (red line) for Tor_NH_N (top panels) and Tor_N_NH (bottom panels). For each protonation state, three replicas were simulated.

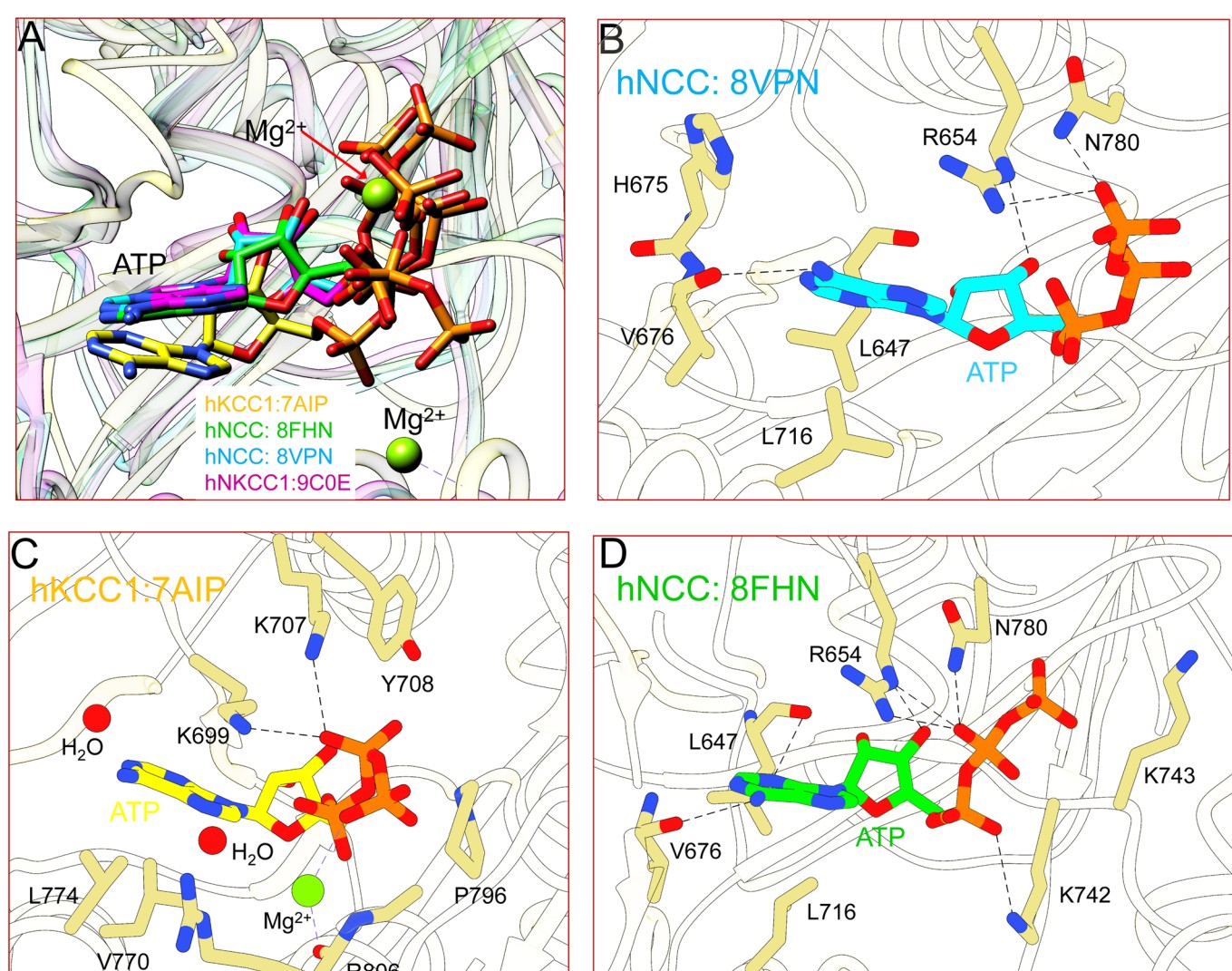

**Figure EV5. Comparison of ATP poses and its coordinating residues in NKCC1, NCC, and KCC1.**

(A) A superimposition of ATP in human KCC1 (PDB code: 7AIP), human NCC (PDB code: 8VPN and 8FHN) and human NKCC1 (PDB code: 9C0E). (B) The ATP and coordinating residues in human NCC (PDB code: 8VPN) were highlighted in sticks. (C) The ATP and coordinating residues in human KCC1 (PDB code: 7AIP) were highlighted in sticks. (D) The ATP and coordinating residues in human NCC (PDB code: 8FHN) were highlighted in sticks.

