## [Peer Review File · The EMBO Journal]

Structural basis for human NKCC1 inhibition by loop diuretic drugs

Erhu Cao, Yongxiang Zhao, Pietro Vidossich, Biff Forbush, Junfeng Ma, Jesse Rinehart, and Marco De Vivo

Corresponding author(s): Erhu Cao (erhu.cao@biochem.utah.edu)

Review Timeline:

Submission Date:	14th Jun 24
Editorial Decision:	24th Jul 24
Revision Received:	17th Oct 24
Editorial Decision:	26th Nov 24
Revision Received:	11th Dec 24
Accepted:	2nd Jan 25

Editor: William Teale

Transaction Report:

Dear Dr. Cao,

Thank you again for the submission of your manuscript entitled "Structural Bases for Inhibition of the NKCC1 transporter by Two Chemical Classes of Loop Diuretic Drugs" (EMBOJ-2024-118205) and for your patience during the review process. We have now received the reports from the referees, which I copy below.

As you can see from the comments, referee #2 points to some important technical aspects of your work that will need to be addressed before your manuscript can be published in The EMBO Journal.

However, based on the overall interest expressed in the reports, I would like to invite you to address the comments of all referees in a revised version of the manuscript. I should add that it is The EMBO Journal policy to allow only a single major round of revision and that it is therefore important to resolve the main concerns at this stage. I believe the concerns of the referees are reasonable and addressable, but please contact me if you have any questions, need further input on the referee comments or if you anticipate any problems in addressing any of their points. If you would like to discuss referee #2's report with me, I am always available over Zoom; just let me know a convenient morning for you. Please, follow the instructions below when preparing your manuscript for resubmission.

I would also like to point out that as a matter of policy, competing manuscripts published during this period will not be taken into consideration in our assessment of the novelty presented by your study ("scooping" protection). We have extended this 'scooping protection policy' beyond the usual 3 month revision timeline to cover the period required for a full revision to address the essential experimental issues. Please contact me if you see a paper with related content published elsewhere to discuss the appropriate course of action.

Again, please contact me at any time during revision if you need any help or have further questions.

Thank you very much again for the opportunity to consider your work for publication. I look forward to your revision.

Best regards,

William

William Teale, Ph.D.
Editor
The EMBO Journal

When submitting your revised manuscript, please carefully review the instructions below and include the following items:

- 1) a .docx formatted version of the manuscript text (including legends for main figures, EV figures and tables). Please make sure that the changes are highlighted to be clearly visible.
- 2) individual production quality figure files as .eps, .tif, .jpg (one file per figure).
- 3) a .docx formatted letter INCLUDING the reviewers' reports and your detailed point-by-point response to their comments. As part of the EMBO Press transparent editorial process, the point-by-point response is part of the Review Process File (RPF), which will be published alongside your paper.
- 4) a complete author checklist, which you can download from our author guidelines ([https://wol-prod-cdn.literatumonline.com/pb-assets/embo-site/Author Checklist%20-%20EMBO%20J-1561436015657.xlsx](https://wol-prod-cdn.literatumonline.com/pb-assets/embo-site/Author%20Checklist%20-%20EMBO%20J-1561436015657.xlsx)). Please insert information in the checklist that is also reflected in the manuscript. The completed author checklist will also be part of the RPF.
- 5) Please note that all corresponding authors are required to supply an ORCID ID for their name upon submission of a revised manuscript.
- 6) We require a 'Data Availability' section after the Materials and Methods. Before submitting your revision, primary datasets produced in this study need to be deposited in an appropriate public database, and the accession numbers and database listed under 'Data Availability'. Please remember to provide a reviewer password if the datasets are not yet public (see

<https://www.embopress.org/page/journal/14602075/authorguide#datadeposition>). If no data deposition in external databases is needed for this paper, please then state in this section: This study includes no data deposited in external repositories. Note that the Data Availability Section is restricted to new primary data that are part of this study.

Note - All links should resolve to a page where the data can be accessed.

8) For data quantification: please specify the name of the statistical test used to generate error bars and P values, the number (n) of independent experiments (specify technical or biological replicates) underlying each data point and the test used to calculate p-values in each figure legend. The figure legends should contain a basic description of n, P and the test applied. Graphs must include a description of the bars and the error bars (s.d., s.e.m.).

9) We would also encourage you to include the source data for figure panels that show essential data. Numerical data can be provided as individual .xls or .csv files (including a tab describing the data). For 'blots' or microscopy, uncropped images should be submitted (using a zip archive or a single pdf per main figure if multiple images need to be supplied for one panel). Additional information on source data and instruction on how to label the files are available at .

10) We replaced Supplementary Information with Expanded View (EV) Figures and Tables that are collapsible/expandable online (see examples in <https://www.embopress.org/doi/10.15252/embj.201695874>). A maximum of 5 EV Figures can be typeset. EV Figures should be cited as 'Figure EV1, Figure EV2" etc. in the text and their respective legends should be included in the main text after the legends of regular figures.

12) Our journal encourages inclusion of *data citations in the reference list* to directly cite datasets that were re-used and obtained from public databases. Data citations in the article text are distinct from normal bibliographical citations and should directly link to the database records from which the data can be accessed. In the main text, data citations are formatted as follows: "Data ref: Smith et al, 2001" or "Data ref: NCBI Sequence Read Archive PRJNA342805, 2017". In the Reference list, data citations must be labeled with "[DATASET]". A data reference must provide the database name, accession number/identifiers and a resolvable link to the landing page from which the data can be accessed at the end of the reference. Further instructions are available at .

Further instructions for preparing your revised manuscript:

At EMBO Press we ask authors to provide source data for the main manuscript figures. Our source data coordinator will contact

you to discuss which figure panels we would need source data for and will also provide you with helpful tips on how to upload and organize the files.

We realize that it is difficult to revise to a specific deadline. In the interest of protecting the conceptual advance provided by the work, we recommend a revision within 3 months (22nd Oct 2024). Please discuss the revision progress ahead of this time with the editor if you require more time to complete the revisions. Use the link below to submit your revision:

Link Unavailable

Referee #1:

The manuscript by Zhao et al describes a highly relevant study of the human sodium potassium chloride transporter 1 (NKCC1) which plays a central role in Cl homeostasis and regulating cell volume and neuronal excitability. The authors utilize an elegant approach of co-expressing WNK1/Mo25/SPAK and human NKCC1 using a bicistronic construct for stable cell line generation which allows them to capture the transporter in a fully activated state with key regulatory sites in the N-terminus phosphorylated. They report cryo-EM structures at (2.5 -2.68 Å resolution) of phospho-activated NKCC1 with three different diuretic drugs (furosemide, bumetanide and torsemide), revealing distinct mechanisms of actions which are relevant for their potency under in vivo ionic conditions. A key finding from this study is that bumetanide and furosemide are potassium dependent antagonists which coordinate a potassium ion bound to the active site via a carboxyl group present in the inhibitory drugs, hence explaining their increased potency in the presence of low concentrations of potassium. By contrast, torsemide binds to NKCC1 in a different mode which expels the potassium from the ion coordinating site. Interestingly, they show that the sulfamoyl group of torsemide is dispensable for the interaction with NKCC1, hence paving the way for the design of sulfur-free loop diuretics - an unmet need for patients with intolerance to sulfur allergies.

Another notable finding of this work is that the structures of fully phosphorylated NKCC1 reveal previously unresolved regions of the regulatory N- and C-terminal domains which are shown to interact and may play a role in controlling the stability of the intracellular gate formed by residues K624 (TM8) and D510 (TM6) via long distance interactions mediated via the highly conserved intracellular loop IL1. This interpretation is supported by robust mutagenesis data where mutations at key interaction sites between NTD/CTD and IL1 result in impaired transport activity measured by fluorometric measurements in cells which stably co-express a chloride-sensing membrane-bound YFP variant. Furthermore, convincing structural and mutagenesis data confirm the positive allosteric role of ATP bound to a conserved hydrophobic pocket in the C-terminal domain and rule out a previously postulated, but highly controversial binding site of loop diuretics to this nucleotide binding site. The interpretation that all loop diuretics act via binding to the orthosteric site in the readily accessible extracellular vestibule of NKCC1 makes sense because the poor cell permeability of these diuretic drugs makes it highly unlikely that these would reach the ATP binding pocket in the cytoplasmic CTD.

The electrostatic potential maps and molecular models are of high quality and support the interpretations by the authors. The study is well-written and nicely places the new findings into relation with previous results and observations. Together, the results from this manuscript represent a significant contribution to the field of CCCs and a major advancement in our understanding of the molecular mechanisms of inhibition by loop diuretics. The work is therefore well suited for publication in the EMBO Journal after addressing the minor points outlined below.

Minor points:

1. It would be useful to include mass spec data for digested NKCC1, showing the exact mass of peptides derived from the N-terminus with phosphorylation sites to confirm that the strategy of co-expressing WNK/SPAK and preventing dephosphorylation by phosphatases was indeed successful.
2. The authors have placed ATP into the density map obtained in the presence of furosemide and state that with the other inhibitors, they see similar densities indicative of nucleotide binding, but they are less well resolved. Can the authors please comment if this is endogenous ATP or if they have added it during purification? Is there an explanation for the difference in density observed between the maps obtained for the three different inhibitors?
3. Main Line 305: Region 230-231 has weak density in the maps provided by the authors and the main chain is traced in a different conformation for the model with furosemide compared to the model with torsemide. Due to the weak density, it is speculative to trace the protein backbone and the interpretation of the structural data is experimentally not very well supported. It does make sense that the pT230 would interact with Arg348 upon phosphorylation but the data is not strong enough to clearly show this interaction unambiguously.
4. Main text, line 203 refers to Figure S8, but Cl ions mentioned in the text are only displayed in Figure S9, so please refer to this instead or in addition to S8.
5. Introduce sub-panels in Figures S8 and S9 for better clarity when referring to different structures shown in the Appendix Figure.
6. Main text, line 354: "Although a mechanistic understanding of how ICL1 and phosphosphorylation of NTD act...". Rephrase to:
7. "Although a mechanistic understanding of how ICL1 and regulatory phosphorylation of NTD act...".
8. Main text, line 528: de novo-> ab initio
9. Main text, line 387, atomic details -> near atomic details
10. Figure 5 D: please indicate in the Figure/Figure legend which pdb was used for showing NKCC1 in the inward facing state which was not solved as part of this study and is shown here to highlight differences between the two conformational states.
11. Figure S13: label key residues in inset picture for better clarity
12. Method section: Add a sentence to specify which software programs were used to create cif files with geometric restraints for bumetanide, furosemide and torsemide and how real space refinement was carried out.

Referee #2:

In this report by Zhao and colleagues, the authors investigate the mechanisms by which the loop diuretics, furosemide, bumetanide, and torsemide inhibit transport by phosphorylated NKCC1. Using cryo-EM, the authors observe that the three inhibitors all occupy the extracellular region of the ion transport pathway, stabilizing the transporter in similar outward facing states in which the cytoplasmic domain is rigidly attached to the TMDs. The domains are stabilized by the phosphorylated NTD, which forms contacts with both domains. In one of the states, the authors observe a bound ATP-Mg in the cytoplasmic domain that occupies an ATP binding site that has been observed in other members of the CCC family. Using mutagenesis, the authors are able to validate many of the structural observations. Altogether, these analyses improve our understanding of NKCC transport and clarify the mechanisms of inhibitor binding. However, there are some significant technical limitations, which are listed below, that need to be addressed before this report would be suitable for publication in EMBO Journal.

Comments:

1. Due to the controversial nature of the inhibitor binding sites in NKCCs, extreme care should be taken in modelling ligands into the density maps. For example, in figure 3B, the ATP is incorrectly modelled in the furosemide-bound structure. The authors should carefully investigate each of the ligands to ensure that they are properly modelled.
2. Continuing from point 1, it is critical that any deficiencies in the modelling or reconstructions should be acknowledged as potential limitations. For example, the angular distribution plots for the furosemide-bound and the torsemide-bound structures indicate the presence of anisotropic data. Similarly, the MolProbity scores and fraction of Ramachandran outliers are quite high given the resolution of the reconstructions. Several residues at critical interfaces, including V302, M303 and L926 in the furosemide-bound are visible, but not modelled correctly. The authors should apply available validation and refinement tools to identify any potential deficiencies in their structures would limit their interpretations and attempt to correct them if possible.
3. As the inhibitors are an important feature of this report, it would be helpful to show the distances between the ligands and protein in Figures 2 and S10 and include a supplemental figure showing the density for the ligands and nearby amino acids, ions and water molecules. It would also be helpful to note the threshold used for depicting the densities in the figure legends.
4. The authors should mention earlier in the text that ATP has been observed bound to the intracellular domains of other CCCs. How similar is the binding site observed in furosemide-bound NKCC1 and which residues are conserved? The authors find that

the R801A mutation diminishes transport but not the K889A or D891A mutations. Can the authors speculate on the differential effects of the mutations? Have mutations of the ATP binding site been observed to influence transport of any other members of the CCC family as the text states that "highlighting a 389 conserved role of ATP in regulating all members of CCCs"?

5. What the origin of the ATP resolved in the density map of the furosemide-bound NKCC1? If it was co-purified, it would suggest a very low off-rate (hours to days) and that ATP may serve a structural, rather than regulatory role. Although the density for the ATP is quite clear (if incorrectly modeled) in the furosemide-bound structure, the densities presented in Figure S12 to support ATP binding in the other structures, and in particular the bumetanide map, is unconvincing? Are there any reasons to suggest why ATP binding would be influenced by the presence of an inhibitor?

Minor Comments:

1. Several unlabeled side chains in the TMD are shown in Figure 1. Which side chains are displayed and what is the rationale for displaying these residues?
2. The authors mention that all three states are highly similar. What is the RMSD between the three states? Are the
3. It is very difficult to see which residues are labeled in Figure 4B. Would it be possible to zoom in on the region of interest to enable a better view of the residues being highlighted?

In the past few months since the initial submission, we have performed new experiments and computational analyses to improve our manuscript. We addressed the reviewers' concerns which are detailed below in our point-to-point responses, and accordingly updated main text with the major changes highlighted in yellow for easy tracking. The major improvements are: 1) we established collaboration with Dr. Junfeng Ma (an expert in proteomics) and used mass spectrometry to confirm that our NKCC1 sample indeed bears a phosphate group in each of all previously reported phosphoacceptor sites when purified from WNK1/SPAK/Mo25 co-expressing HEK293 cells; 2) we have corrected errors in our previous structure models and the newly deposited PDB coordinates have better MolProbity scores. We also calculated directional resolutions of our maps to mitigate concerns of uneven angular distribution of our datasets; and 3) we validated furosemide pose by molecular dynamics simulations.

Referee #1:

The manuscript by Zhao et al describes a highly relevant study of the human sodium potassium chloride transporter 1 (NKCC1) which plays a central role in Cl homeostasis and regulating cell volume and neuronal excitability. The authors utilize an elegant approach of co-expressing WNK1/Mo25/SPAK and human NKCC1 using a bicistronic construct for stable cell line generation which allows them to capture the transporter in a fully activated state with key regulatory sites in the N-terminus phosphorylated. They report cryo-EM structures at (2.5 -2.68 Å resolution) of phospho-activated NKCC1 with three different diuretic drugs (furosemide, bumetanide and torsemide), revealing distinct mechanisms of actions which are relevant for their potency under in vivo ionic conditions. A key finding from this study is that bumetanide and furosemide are potassium dependent antagonists which coordinate a potassium ion bound to the active site via a carboxyl group present in the inhibitory drugs, hence explaining their increased potency in the presence of low concentrations of potassium. By contrast, torsemide binds to NKCC1 in a different mode which expels the potassium from the ion coordinating site. Interestingly, they show that the sulfamoyl group of torsemide is dispensable for the interaction with NKCC1, hence paving the way for the design of sulfur-free loop diuretics - an unmet need for patients with intolerance to sulfur allergies.

Another notable finding of this work is that the structures of fully phosphorylated NKCC1 reveal previously unresolved regions of the regulatory N- and C-terminal domains which are shown to interact and may play a role in controlling the stability of the intracellular gate formed by residues K624 (TM8) and D510 (TM6) via long distance interactions mediated via the highly conserved intracellular loop IL1. This interpretation is supported by robust mutagenesis data where mutations at key interaction sites between NTD/CTD and IL1 result in impaired transport activity measured by fluorometric measurements in cells which stably co-express a chloride-sensing membrane-bound YFP variant. Furthermore, convincing structural and mutagenesis data confirm the positive allosteric role of ATP bound to a conserved hydrophobic pocket in the C-terminal domain and rule out a previously postulated, but highly controversial binding site of loop diuretics to this nucleotide binding site. The interpretation that all loop diuretics act via binding to the orthosteric site in the readily accessible extracellular vestibule of NKCC1 makes sense because the poor cell permeability of these diuretic drugs makes it highly unlikely that

these would reach the ATP binding pocket in the cytoplasmic CTD.

The electrostatic potential maps and molecular models are of high quality and support the interpretations by the authors. The study is well-written and nicely places the new findings into relation with previous results and observations. Together, the results from this manuscript represent a significant contribution to the field of CCCs and a major advancement in our understanding of the molecular mechanisms of inhibition by loop diuretics. The work is therefore well suited for publication in the EMBO Journal after addressing the minor points outlined below.

We appreciate the reviewer #1 for his/her positive assessment of our work and constructive comments that have helped to improve our manuscript.

Minor points:

1. It would be useful to include mass spec data for digested NKCC1, showing the exact mass of peptides derived from the N-terminus with phosphorylation sites to confirm that the strategy of co-expressing WNK/SPAK and preventing dephosphorylation by phosphatases was indeed successful.

Agreed. We have established collaboration with Dr. Junfeng Ma whose group has performed mass spectrometry analyses of our NKCC1 sample purified from kinases co-expressing HEK293 cells. The mass spectra showed that Thr203, Thr212, Thr217, and Thr230 residues are phosphorylated with almost 100% confidence (see **Appendix Figure S1**).

2. The authors have placed ATP into the density map obtained in the presence of furosemide and state that with the other inhibitors, they see similar densities indicative of nucleotide binding, but they are less well resolved. Can the authors please comment if this is endogenous ATP or if they have added it during purification? Is there an explanation for the difference in density observed between the maps obtained for the three different inhibitors?

Thanks for pointing out this important omission of key experimental details in our original manuscript. When preparing pNKCC1/bumetanide and pNKCC1/torseamide samples, we didn't supply ATP during purification because we were negligent of CCCs regulation by ATP at the time. The poorly defined densities in the CTD cavity of those two maps are difficult to assign a ligand, **but they definitely cannot be accounted for by the loop diuretic drugs in the samples**. When it dawned on us that ATP could be a fundamental modulator of CCCs, we added 100 μ M ATP throughout all steps of pNKCC1/furosemide purification. The resulting pNKCC1/furosemide map enabled modelling of an ATP-Mg²⁺ molecule into a well-resolved density. We have included these details in the updated main text (**Line253-265**).

3. Main Line 305: Region 230-231 has weak density in the maps provided by the authors and the main chain is traced in a different conformation for the model with furosemide compared to the model with torseamide. Due to the weak density, it is speculative to trace the protein backbone and the interpretation of the structural data is experimentally not very well supported. It does make sense that the pT230 would interact with Arg348 upon phosphorylation but the data is not strong enough to clearly show this interaction unambiguously.

We agreed with the reviewer that pT230 density is not as clear-cut as those of pT212 and pT217 possible because T230 resides in a flexible loop. In the updated main text, we now tuned down our statement by acknowledging that T230 can only be tentatively modeled with a phosphate group (**Line 319-320**). The reason that we only observed pT230 in pNKCC1/furosemide, but not in pNKCC1/bumetanide and pNKCC1/torseamide maps is because we became more skillful when we prepared the pNKCC1/furosemide sample. We first determined pNKCC1/bumetanide and pNKCC1/torseamide structures using the WNK1-SPAK-Mo25 co-expressing system combined with the treatment with a kinase-activating Cl⁻ free buffer. However, the same strategy failed to determine a pNKCC1/furosemide structure. To overcome this obstacle, we used human NKCC1 A492E mutant which exhibits increased binding affinity for loop diuretics, and further supplemented the phosphatases inhibitor calyculin A to block unwanted NKCC1 dephosphorylation during purification. We suspect that pNKCC1/furosemide is more completely phosphorylated than in pNKCC1/bumetanide and pNKCC1/torseamide samples, enabling us to observe pT230 (albeit less clear than pT217 and pT212) and to trace more residues preceding pT212 up to T203. We thus used the map and model of pNKCC1/furosemide to discuss NKCC1 phosphoregulatory mechanism.

We feel confident to propose that allosteric communication between TMDs and cytosolic domains of NKCC1 fostered by pT230-R348 (and other interactions) promotes NKCC1 activation because an analogous pSer73-Lys196 interaction exists in NCC as reported in our recent work (**PMID: 39143061**). We do agree with the reviewer #1 that a clearer map in this key region would strengthen our hypothesis. We (and others) will certainly continue to explore N(K)CCs phosphoregulation in the future, especially to elucidate whether and how the three isoforms of NKCC2 differ in their response to kinase activation (see **lines 438 -443** in the discussion section).

4. Main text, line 203 refers to Figure S8, but Cl ions mentioned in the text are only displayed in Figure S9, so please refer to this instead or in addition to S8.

Thanks for the suggestion. This is now done.

5. Introduce sub-panels in Figures S8 and S9 for better clarity when referring to different structures shown in the Appendix Figure.

Thanks for the suggestion. This is now done.

6. Main text, line 354: "Although a mechanistic understanding of how ICL1 and phosphorylation of NTD act...". Rephrase to:

7. "Although a mechanistic understanding of how ICL1 and regulatory phosphorylation of NTD act...".

Thanks for the suggestion. We changed the text (**Line 374-376**).

8. Main text, line 528: de novo-> ab initio

Thanks for the suggestion. We changed the text.

9. Main text, line 387, atomic details -> near atomic details

Thanks for the suggestion. We changed the text to be more accurate.

10. Figure 5 D: please indicate in the Figure/Figure legend which pdb was used for showing NKCC1 in the inward facing state which was not solved as part of this study and is shown here to highlight differences between the two conformational states.

Thanks for the suggestion. We indicated PDB code in Figure legends.

11. Figure S13: label key residues in inset picture for better clarity

Thanks for the suggestion. This is done (see Appendix Figure S11).

12. Method section: Add a sentence to specify which software programs were used to create cif files with geometric restraints for bumetanide, furosemide and torsemide and how real space refinement was carried out.

Thanks for the suggestions. Overall, our model building and refinement procedures are of standard practice. All cif files for ligands (loop diuretic drugs and ATP) were generated in Coot using the ligand builder functionality; models were iteratively refined in PHENIX with the morphing and simulated annealing functionalities opted in in early stages but with only default geometry constraints applied in the later stages. We have now included these details in the updated main text (**Lines 604-610**).

Referee #2:

In this report by Zhao and colleagues, the authors investigate the mechanisms by which the loop diuretics, furosemide, bumetanide, and torsemide inhibit transport by phosphorylated NKCC1. Using cryo-EM, the authors observe that the three inhibitors all occupy the extracellular region of the ion transport pathway, stabilizing the transporter in similar outward facing states in which the cytoplasmic domain is rigidly attached to the TMDs. The domains are stabilized by the phosphorylated NTD, which forms contacts with both domains. In one of the states, the authors observe a bound ATP-Mg in the cytoplasmic domain that occupies an ATP binding site that has been observed in other members of the CCC family. Using mutagenesis, the authors are able to validate many of the structural observations. Altogether, these analyses improve our understanding of NKCC transport and clarify the mechanisms of inhibitor binding. However, there are some significant technical limitations, which are listed below, that need to be addressed before this report would be suitable for publication in EMBO Journal.

We greatly appreciate the reviewer #2 for pointing out the limitations (and omissions) of our manuscript for improvement.

Comments:

1. Due to the controversial nature of the inhibitor binding sites in NKCCs, extreme care should

be taken in modelling ligands into the density maps. For example, in figure 3B, the ATP is incorrectly modelled in the furosemide-bound structure. The authors should carefully investigate each of the ligands to ensure that they are properly modelled.

Thanks for pointing out this embarrassing mistake. We actually also realized this error shortly after the initial submission. We now corrected the chiral error of ATP in the coordinate (see **validation reports**). We also double-checked models of diuretic drugs in maps and validated their experimental poses using molecular dynamics simulations (**Figure EV2**).

2. Continuing from point 1, it is critical that any deficiencies in the modelling or reconstructions should be acknowledged as potential limitations. For example, the angular distribution plots for the furosemide-bound and the torsemide-bound structures indicate the presence of anisotropic data. Similarly, the MolProbity scores and fraction of Ramachandran outliers are quite high given the resolution of the reconstructions. Several residues at critical interfaces, including V302, M303 and L926 in the furosemide-bound are visible, but not modelled correctly. The authors should apply available validation and refinement tools to identify any potential deficiencies in their structures would limit their interpretations and attempt to correct them if possible.

Thank the reviewer for pointing out these errors which have now been corrected. To improve our models, we first used PHENIX local sharpening tool to enhance our three NKCC1/loop diuretic drug maps, which have been deposited together with the unsharpened raw maps in EMDB. We corrected all the errors using these PHENIX sharpened maps in Coot and then iteratively refined the resulting models against the sharpened maps in PHENIX in real space. The updated models have no Ramachandran outliers and their MolProbity scores are improved, especially for the NKCC1/torsemide model which has improved from 3.88 to 2.72.

We agree with the reviewer that NKCC1 assumes preferred orientations in our datasets as often occurs in many other cryo-EM samples. To mitigate this concern, we used a remote 3DFSC server to calculate sphericity and directional resolutions of our three maps (see **Appendix Figures 3F, 5F, and 7F**). After such computational analyses and visual inspection of our maps, we feel that uneven angular distribution is not severe enough to impact our interpretations and conclusions drawn from these maps as they all have good sphericity.

3. As the inhibitors are an important feature of this report, it would be helpful to show the distances between the ligands and protein in Figures 2 and S10 and include a supplemental figure showing the density for the ligands and nearby amino acids, ions and water molecules. It would also be helpful to note the threshold used for depicting the densities in the figure legends.

Thanks for the suggestions. These are now done (see **Figures 2, EV3, and EV4**).

4. The authors should mention earlier in the text that ATP has been observed bound to the intracellular domains of other CCCs. How similar is the binding site observed in furosemide-bound NKCC1 and which residues are conserved? The authors find that the R801A mutation diminishes transport but not the K889A or D891A mutations. Can the authors speculate on the

differential effects of the mutations? Have mutations of the ATP binding site been observed to influence transport of any other members of the CCC family as the text states that "highlighting a 389 conserved role of ATP in regulating all members of CCCs"?

Thanks for the suggestions. As also mentioned in our response to the question #2 raised by the reviewer #1, we have added new texts to discuss ATP binding sites in NKCC1, NCC, and KCC1 (Lines 253-265, 289-293, and 296-305). Although ATP poses are very similar when nestled in their respective CTD pocket of CCCs, ATP binding residues are not strictly conserved in these transporters except for they all using basic residue(s) (i.e., Arg or Lys) to coordinate ATP (Figure EV5). In human NKCC1, R801 uses its sidechain to extensively engage with the adenine, ribose, and phosphate of ATP via cation- π and polar interactions (Figure 3C), abolishing those interactions by R801A would significantly diminish ion flux activity. Other residues such as K889 and D891 do not engage in such extensive interactions with ATP and, in many cases, use their mainchain oxygen or nitrogen atoms for ATP coordination (Figure 3C). These may explain why we did not see significant effect of K889A or D891A on NKCC1 ion transport rate. These observations are also consistent with our recent NCC study (PMID: 39143061) in which we found that NCC R654A mutant reduces ion flux rate, but other mutations do not notably affect NCC function. Since we are yet to explore ATP regulation of other CCC members, we now state that CCCs are all *potentially* regulated by ATP (line 412).

5. What is the origin of the ATP resolved in the density map of the furosemide-bound NKCC1? If it was co-purified, it would suggest a very low off-rate (hours to days) and that ATP may serve a structural, rather than regulatory role. Although the density for the ATP is quite clear (if incorrectly modeled) in the furosemide-bound structure, the densities presented in Figure S12 to support ATP binding in the other structures, and in particular the bumetanide map, is unconvincing? Are there any reasons to suggest why ATP binding would be influenced by the presence of an inhibitor?

As explained in our response to the Reviewer #1 question #2, we didn't add extra ATP in the pNKCC1/bumetanide and pNKCC1/torseamide samples, but included 100 μ M ATP throughout all purification steps when preparing the pNKCC1/furosemide complex. We believe that endogenous ATP was washed away during purification due high off-rate, which may explain poorly defined density in the ATP binding pocket when exogenous ATP was not supplied. We are not aware of evidence that loop diuretics affect ATP binding.

Minor Comments:

1. Several unlabeled side chains in the TMD are shown in Figure 1. Which side chains are displayed and what is the rationale for displaying these residues?

We have now corrected this mistake.

2. The authors mention that all three states are highly similar. What is the RMSD between the three states?

We have now done superimposition analyses (see **Appendix Figure S13**). The average backbone RMSD between these structures is 0.9 Å.

3. It is very difficult to see which residues are labeled in Figure 4B. Would it be possible to zoom in on the region of interest to enable a better view of the residues being highlighted?

Thanks for the suggestion. This is now done (see **Figure 4A**).

Dear Dr. Cao,

Thank you for addressing the reviewers' comments. As you will see below, you have addressed all concerns satisfactorily.

Before I can finally accept the manuscript, there are some remaining editorial points which need to be addressed. In this regard, would you please:

- acknowledge funding in the manuscript "Acknowledgements" section,
- include up to five keywords,
- rename the data and materials availability section 'Data Availability',
- rename the competing interests section 'Disclosure and competing interests statement',
- remove the author credit section from the manuscript,
- include a callout for Figure 1A-F and ensure that Figures 2 and 3 are called out in the correct order in the manuscript,
- complete the author checklist in full,
- upload the appendix file as a PDF, filling in page numbers in the table of contents, and changing the subtitle "Supplementary Materials for" to "Appendix for",
- label individual source data figure panels in the corresponding zip files (i.e. in a zip folder containing source data files for Fig. 1 there should be individual folders named 1B, 1C, 1D, 1E... with the corresponding files, the folders can also be named 1B-D, or something similar); there should be clearly labeled 'panel level folders' for all figures,
- ensure all data files in external repositories are made public upon final acceptance of this work,
- provide specific URLs for Electron Microscopy Data Bank (EMDB) (EMD-45081, EMD-45083, and EMD-45084 683) and Protein Data Bank (PDB) (9C0E, 9C0G, and 9C0H) datasets in the data availability statement,
- ensure that legends for figure panels 2a-c are provided in a sequential manner (i.e., the legend for figure 2b should be provided before legend of figure 2c),
- provide figure legends for figure EV 3a-c in the manuscript,
- state exact p values in the legends of figures 3d; 4c; 5e,
- define the measure of centre used in the legends of figures 2e-f; EV 1d, and
- correct the section order as follows: Title page - Abstract & Keywords - Introduction - Results - Discussion - Methods - Data Availability - Acknowledgements - Disclosure and Competing Interests Statement - References - Figure Legends - Table(s) - Expanded View Figure Legends.

We include a synopsis of the paper (see <http://emboj.embojpress.org/>). Please provide me with a general summary image, a two-sentence summary statement and 3-5 bullet points that capture the key findings of the paper.

I look forward to receiving these changes. EMBO Press is an editorially independent publishing platform for the development of EMBO scientific publications.

Best wishes,

William Teale

William Teale, PhD
Editor
The EMBO Journal
w.teale@embojournal.org

See also figure legend guidelines: <https://www.embojpress.org/page/journal/14602075/authorguide#figureformat>

- a point-by-point response to the referees' comments, with a detailed description of the changes made (as a word file).
- a word file of the manuscript text.
- individual production quality figure files (one file per figure)

- a complete author checklist, which you can download from our author guidelines (<https://www.embopress.org/page/journal/14602075/authorguide>).

- Expanded View files (replacing Supplementary Information)

We realize that it is difficult to revise to a specific deadline. In the interest of protecting the conceptual advance provided by the work, we recommend a revision within 3 months (24th Feb 2025). Please discuss the revision progress ahead of this time with the editor if you require more time to complete the revisions. Use the link below to submit your revision:

Link Unavailable

Referee #1:

The concerns and limitations raised in the first round of reviews have been addressed and the revised manuscript by Zhao et al. is suitable for a publication in the EMBO Journal.

Referee #2:

The authors have addressed all of my concerns and the manuscript is suitable for publication.

The authors have addressed all minor editorial requests.

Dear Dr. Cao,

I am pleased to inform you that your manuscript has been accepted for publication in the EMBO Journal.

Congratulations to you and your team on some really elegant structures!

Best wishes,

William Teale

William Teale, PhD
Editor
The EMBO Journal
w.teale@embojournal.org

** Click here to be directed to your login page: *Link Unavailable*